# Calcium dynamics regulating the timing of decision-making in *C. elegans*

Yuki Tanimoto[1†], Akiko Yamazoe-Umemoto[1†], Kosuke Fujita[1‡], Yuya Kawazoe[1], Yosuke Miyanishi[1], Shuhei J Yamazaki[1], Xianfeng Fei[2], Karl Emanuel Busch[3], Keiko Gengyo-Ando[4], Junichi Nakai[4], Yuichi Iino[5], Yuishi Iwasaki[6], Koichi Hashimoto[7], Koutarou D Kimura[1*]

[1]Department of Biological Sciences, Graduate School of Science, Osaka University, Toyonaka, Japan; [2]Faculty of Science and Technology, Tohoku Bunka Gakuen University, Sendai, Japan; [3]Centre for Integrative Physiology, The University of Edinburgh, Edinburgh, United Kingdom; [4]Graduate Shool of Science and Engineering, Brain and Body System Science Institute, Saitama University, Saitama, Japan; [5]Department of Biological Sciences, Graduate School of Science, The University of Tokyo, Tokyo, Japan; [6]Department of Intelligent Systems Engineering, Ibaraki University, Hitachi, Japan; [7]Graduate School of Information Sciences, Tohoku University, Sendai, Japan

**\*For correspondence:** kokimura-lab@umin.ac.jp

[†]These authors contributed equally to this work

**Present address:** [‡]Department of Ophthalmology, Graduate School of Medicine, Tohoku University, Sendai, Japan

**Competing interests:** The authors declare that no competing interests exist.

**Abstract** Brains regulate behavioral responses with distinct timings. Here we investigate the cellular and molecular mechanisms underlying the timing of decision-making during olfactory navigation in *Caenorhabditis elegans*. We find that, based on subtle changes in odor concentrations, the animals appear to choose the appropriate migratory direction from multiple trials as a form of behavioral decision-making. Through optophysiological, mathematical and genetic analyses of neural activity under virtual odor gradients, we further find that odor concentration information is temporally integrated for a decision by a gradual increase in intracellular calcium concentration ($[Ca^{2+}]_i$), which occurs via L-type voltage-gated calcium channels in a pair of olfactory neurons. In contrast, for a reflex-like behavioral response, $[Ca^{2+}]_i$ rapidly increases via multiple types of calcium channels in a pair of nociceptive neurons. Thus, the timing of neuronal responses is determined by cell type-dependent involvement of calcium channels, which may serve as a cellular basis for decision-making.

## Introduction

Brains process sensory information to generate various kinds of physiological responses with different timings (*i.e.*, with different latencies to respond): For example, motor control, foraging, decision-making and the sleep-wake cycle range on timescales from milliseconds to days (*Buhusi and Meck, 2005*; *Richelle and Lejeune, 1980*). In decision-making, animals choose one from multiple behavioral options based on environmental sensory information, where a temporal delay is associated with the certainty of sensory information. In primates and rodents, increases in neural activity during the delay period according to the sensory information ('evidence accumulation') has been described as a key physiological basis for the timing of decision-making (*Carandini and Churchland, 2013*; *Gold and Shadlen, 2007*; *Schall, 2001*; *Shadlen and Newsome, 2001*). For example, clear sensory information causes a faster rise in neuronal firing rate to a threshold and faster behavioral choice, whereas uncertain information causes a slower rise in the firing rate and slower behavioral choice. Despite their essential roles in the timing of decision-making, the neural mechanisms that generate

**eLife digest** Animals use information from their environment to make decisions, like where to go, what to eat, and with whom to mate. This information may be changing or confusing, and decisions may be quick when the sensory information is clear, or slower when the sensory clues are muddled. Scientists often study this kind of decision-making in monkeys and rodents, but it can be hard to pinpoint the exact decision-making mechanisms because these animals have hundreds of millions neurons in their brains.

Studying the mechanisms that underlie decision-making can be easier in a simpler organism with fewer neurons. A tiny roundworm called *Caenorhabditis elegans* is one such creature, with only 302 neurons. These worms avoid noxious odors, by first wandering around when they detect the odor, and then fleeing. About 80% of the time the worms flee in the correct direction to escape the foul smell. However, it was not clear how the worms decided which direction to flee.

Now, Tanimoto, Yamazoe-Umemoto et al. show that the worms choose which direction to move by mathematically calculating information about odor concentrations. In the experiments, a robotic microscope simultaneously measured nerve activity and the worm's behavior while an odor was presented. Specifically, the amount of calcium in the neurons was measured. The experiments showed that when the worms were wandering to determine which direction to flee the amount of calcium in the neurons changed in proportion to how much the concentration of the odor changed overtime.

The experiments suggest that the animals use a mathematical process called integration to add up the changes in the concentration of the odor over time, and when the total reaches a certain threshold the animal successfully moves away from the source. Tanimoto, Yamazoe-Umemoto et al. also identified the gene that enables these calculations. More complicated animals make similar calculations that take into account environmental changes over time when making a decision. Future experiments are needed to determine if more complex animals also use the same mechanism as *C. elegans*, and whether the same gene is responsible.

evidence accumulation still need to be clarified. Theoretical studies suggest that evidence accumulation is mediated by recurrent neural circuits (*Gold and Shadlen, 2007*; *Wang, 2008*) while intracellular mechanisms, such as calcium signaling via N-methyl-D-aspartate (NMDA) receptors, calcium-activated nonspecific cation (CAN) channels, and/or voltage-gated calcium channels (VGCCs), have also been proposed (*Curtis and Lee, 2010*; *Major and Tank, 2004*).

The cellular and molecular bases of simple decision-making have been studied in invertebrate animals because of the simplicity and accessibility of their nervous systems (*Kristan, 2008*). For example, a neuron that biases decisions was identified based on optical monitoring of the neuronal activities in the medical leech (*Briggman et al., 2005*), and genetic analyses identified a neuropeptide and a catecholamine receptor that underpin simple decision-making tasks in *Drosophila melanogaster* and in *Caenorhabditis elegans*, respectively (*Bendesky et al., 2011*; *Yang et al., 2008*). In addition, a recent behavioral study showed that *Drosophila* temporally accumulates sensory evidence for decision-making (*DasGupta et al., 2014*). However, the physiological mechanisms of these decision-making tasks have not been elucidated, and thus not discussed in terms of their possible commonalities with the mechanisms in mammals.

In this study, we reveal the cellular and molecular mechanisms of the timing of decision-making in *C. elegans*. We first show that the animals migrate in an appropriate direction with unexpectedly high efficiency, likely based on the gradient of a repulsive odorant. From simultaneous monitoring of behavior and neural activity in virtual odor gradients, we find that two pairs of sensory neurons regulate this behavioral response in an opposing manner with different temporal dynamics. A pair of ASH nociceptive neurons exhibits a time-differential-like response to an increase in the odor concentration, which leads to a bout of turns in random directions similar to a 'reflex'. In contrast, a pair of AWB olfactory neurons exhibits a time-integral response to a decrease in the odor concentration, which leads to turn suppression with a temporal delay resembling 'deliberation'. The AWB response is mediated by a gradual calcium influx mainly via L-type VGCCs whereas the ASH response is

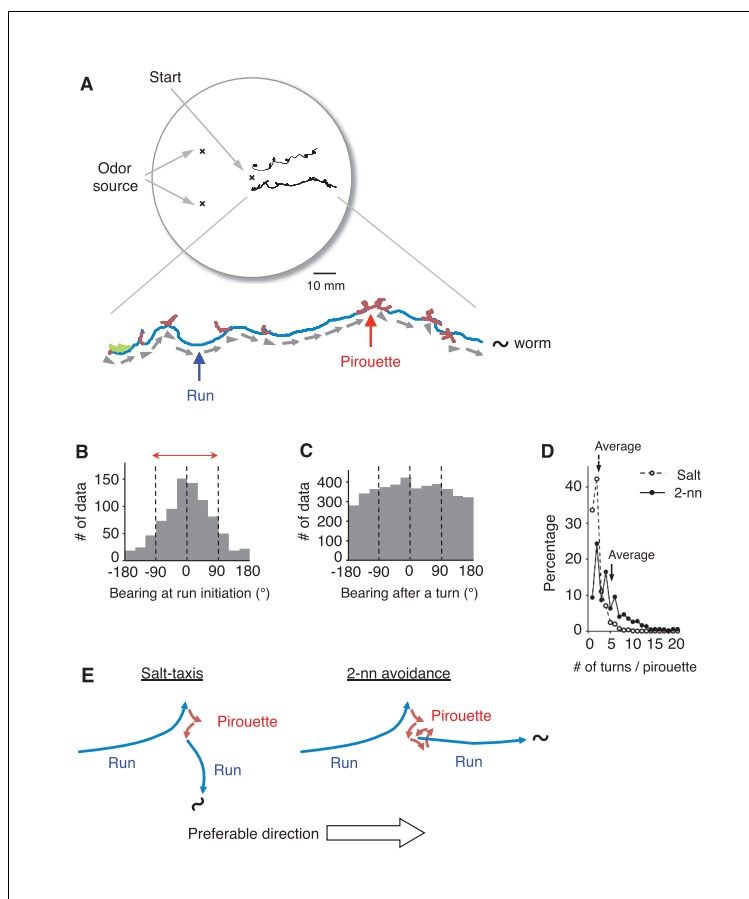

**Figure 1.** *C.elegans* selectively initiates runs away from the odor source. (**A**) Examples of the tracks of 2 animals during 12 min of 2-nonanone avoidance assay, overlaid on a schematic drawing of a 9 cm plate. One of the tracks is magnified below. In the magnified view, pirouettes are red and runs are blue. Arrow heads and arrows indicate the directions of run initiation and those during runs, respectively. (**B**) Histogram indicates the bearings at run initiation during 2-nonanone avoidance (*i.e.*, the bearing of the arrow heads in panel A, and the initial bearing of the blue arrows in panel E). The bearing was determined as $B = 0°$ when animals migrated directly away from the odor source (= down the gradient) and ±180° when they migrated directly toward the source (= up the gradient). Migration away from the odor source (*i.e.*, within ±90° bearings; red arrow) comprised 78.4% of all data. (**C**) Histogram indicates all the bearings after a turn during pirouettes, including those that later switched to runs (*i.e.*, the initial bearings of the red plus blue arrows in panel **E**). (**D**) Percentages of turn numbers per pirouette during 2-nonanone avoidance (solid line and filled circles) and in salt-taxis (dashed lines and open circles). The average number of turns in a pirouette was significantly larger during 2-nonanone avoidance than in salt-taxis (5.2 vs. 2.2 indicated by arrows, respectively; p<0.001 by Mann-Whitney test). (**E**) Schematic drawings of salt-taxis and 2-nonanone avoidance. Typical chemotaxis such as salt-taxis in *C. elegans* is regulated by the pirouette (*i.e.*, biased random walk) strategy, where the animal initiates runs mostly in a random direction after a pirouette (***Pierce-Shimomura et al., 1999***). In contrast, in 2-nonanone avoidance, an appropriate migratory direction is chosen from multiple trials in a pirouette. We refer to this as the 'pivot-and-go' strategy. All the statistical details are shown in ***Supplementary file 1***. For panels B, C and D (2-nonanone avoidance), the data are from 100 wild-type animals. Panel D (salt-taxis) is from 64 wild-type animals. The following figure supplement is available for ***Figure 1***.

The following figure supplement is available for figure 1:

**Figure supplement 1.** Differences between 2-nonanone avoidance behavior and salt-taxis of *C.elegans*.

---

mediated by a rapid calcium influx via multiple types of calcium channels. Thus, our results indicate that the timing of a sensory response, such as deliberate decision-making or rapid reflex, is determined by cell type-dependent involvement of calcium channels.

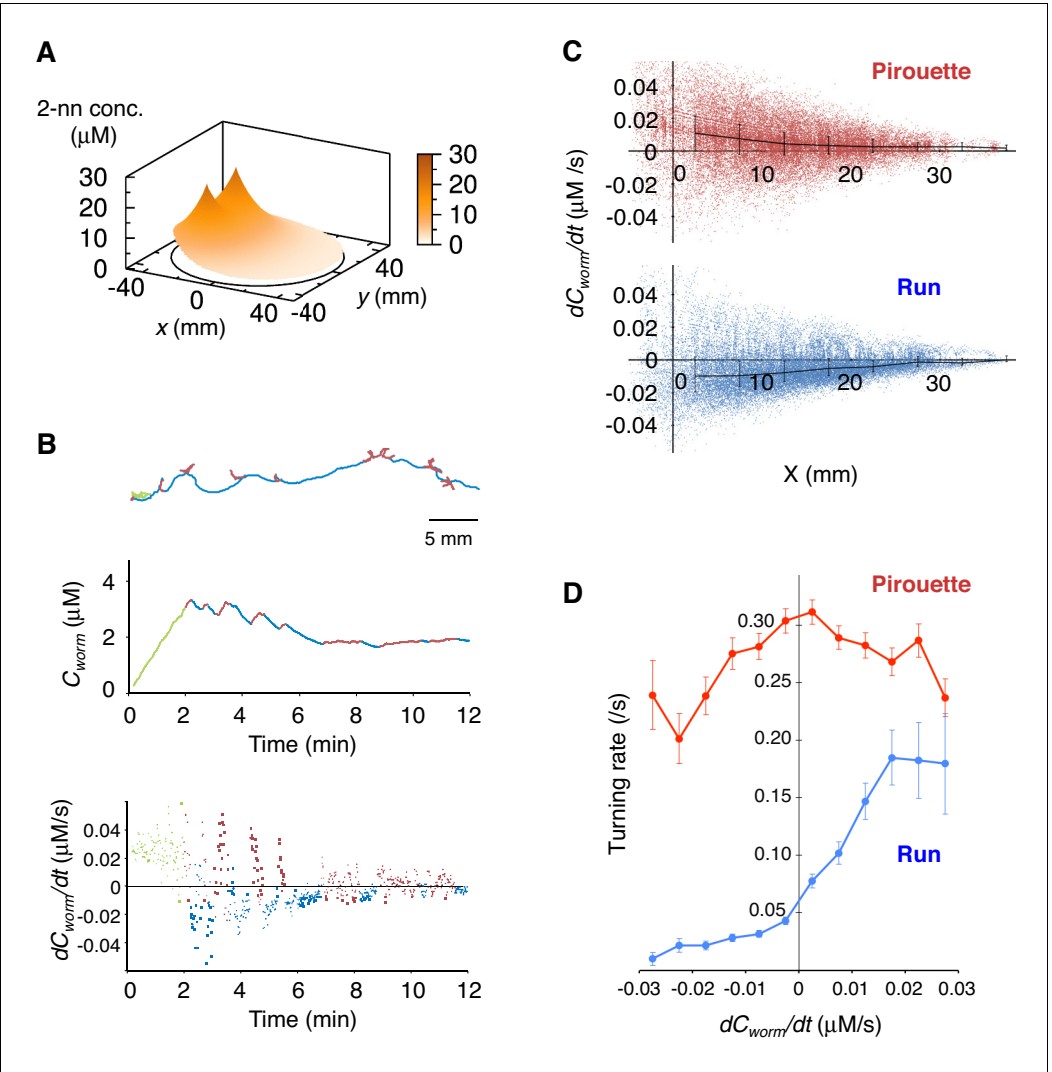

**Figure 2.** Pirouettes and runs are distinct behavioral states, which are associated with positive and negative $dC_{worm}/dt$, respectively. (A) Fitted odor gradient over the assay plate at 12 min, based on the actual measurements shown in *Figure 2—figure supplement 1D*. (B) (Top) Same with the magnified view of an animal's trajectory in *Figure 1A*. (Middle, bottom) Graphs showing the 2-nonanone concentration ($C_{worm}$: middle) or temporal changes in it ($dC_{worm}/dt$: bottom) at this animal's position at each second during the odor avoidance behavior. As in *Figure 1A*, pirouettes and runs are red and blue, respectively. Most of the animals did not migrate much during the first 2 min and were excluded from the analysis (green) because of the rapid increases in the odor concentration during this period. See also *Figure 2—figure supplement 1E* for another example. (C) Correlation between $dC_{worm}/dt$ and pirouettes or runs. $dC_{worm}/dt$ of 2-nonanone was plotted against the animal's x position for each second during pirouettes (top) or runs (bottom). The bars represent the median ± quartiles for each 5 mm fraction. (D) The responsiveness to the instantaneous $dC_{worm}/dt$ differed between pirouettes and runs. The turning rate was determined as the relationship between $dC_{worm}/dt$ during one second of migration and the probability of turning in the next second. Average turning rates ± SEM for every 0.005 µM/s bin during pirouettes (red line) or runs (blue line) are shown. The data in panels C and D are from the same 100 wild-type animals as in *Figure 1*. The following figure supplement is available for *Figure 2*.

The following figure supplement is available for figure 2:

**Figure supplement 1.** Measurement of the gaseous 2-nonanone gradient in the plate assay paradigm.

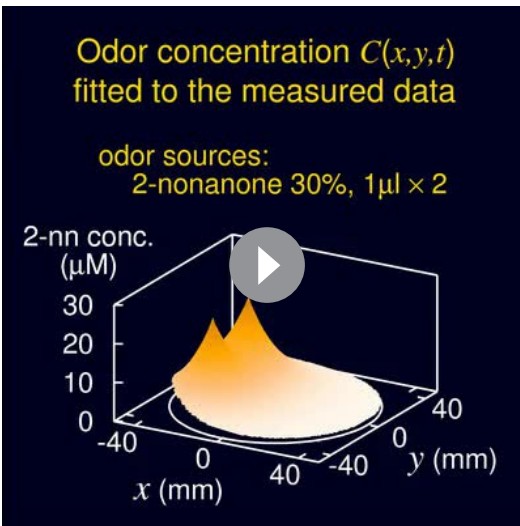

**Video 1.** Time-course changes in the fitted 2-nonanone concentration.      Although the odor sources were two circles of ~5 mm diameter in the real experiment, they were treated as points in the simulation.

## Results

### Efficient directional choice in odor avoidance behavior

*C. elegans* avoids the odorant 2-nonanone, and this odor avoidance behavior is regulated by stimulus-dependent transitions between the two behavioral states, 'pirouette' (a period of short migrations divided by turns) and 'run' (a period of long straight migration) (*Figure 1A* and *Figure 1—figure supplement 1A*) (*Bargmann et al., 1993*; *Kimura et al., 2010*). The pirouette strategy (a form of 'biased random walk') is the major behavioral strategy used by these animals for chemotaxis and thermotaxis, and the choice of migratory direction at run initiation is considered to be random in this strategy (*Lockery, 2011*; *Pierce-Shimomura et al., 1999*). We found, however, that *C. elegans* appropriately chose the migratory direction in 2-nonanone avoidance: 78.4% of the migratory directions at run initiation and 83.5% during runs were away from the odor source (*Figure 1B* and *Figure 1—figure supplement 1B*). Thus, during odor avoidance, animals chose the appropriate direction about four times

more frequently than the inappropriate direction (~80% versus~20%). This probability of run initiation in the appropriate direction for odor avoidance was far higher than that in salt-taxis, the best-studied chemosensory behavior of the animal (59.3% in *Figure 1—figure supplement 1C and D*, and ~56% in *Pierce-Shimomura et al., 1999*), and the probability exceeded, or was at least comparable to, that of odor-taxis in *Drosophila* larvae (73.9%) (*Gomez-Marin et al., 2011*; *Louis et al., 2008*). Appropriate directions were chosen from multiple exploratory short migrations in random directions during pirouettes (*Figure 1C and D*). These results indicate that *C. elegans* can efficiently choose the appropriate migratory direction on a repulsive odor gradient (*Figure 1E*).

### *C. elegans* regulates its behavioral state based on subtle increases and decreases in odor concentration

Next, to understand the correlations between sensory information and appropriate directional choice, we developed a method for measuring and determining the dynamic spatio-temporal pattern of the naturally evaporating and diffusing odor gradient (*Figure 2A* and *Figure 2—figure supplement 1A–D*, and *Video 1*). The measured odor gradient was then used to calculate the odor concentrations that each animal experienced at each position at every second ($C_{worm}$ in the middle panels of *Figure 2B* and *Figure 2—figure supplement 1E*). We found that the temporal changes in $C_{worm}$ ($dC_{worm}/dt$) were strongly correlated with the two behavioral states of the animals (the bottom panels of *Figure 2B* and *Figure 2—figure supplement 1E*, and *Figure 2C*). During pirouettes, the $dC_{worm}/dt$ values were mostly positive because the animals did not migrate much while the odor concentration on the plate was increasing due to sustained evaporation of the odor from the source (*Figure 2—figure supplement 1D* and *Video 1*). During runs, in contrast, $dC_{worm}/dt$ values were mostly negative because the animals migrated down the gradient. These correlations may suggest that pirouettes and runs are caused by positive and negative $dC_{worm}/dt$, respectively. In addition, we further found that the relationships between animals' responsiveness and instantaneous $dC_{worm}/dt$ largely differed between pirouettes and runs (*Figure 2D*), suggesting that pirouettes and runs are physiologically distinct behavioral states in terms of sensory response. Taken together, these results suggest that the efficient transitions between discrete behavioral states based on odor concentration information may lead to the appropriate choice of migratory direction as a simple form of decision-making.

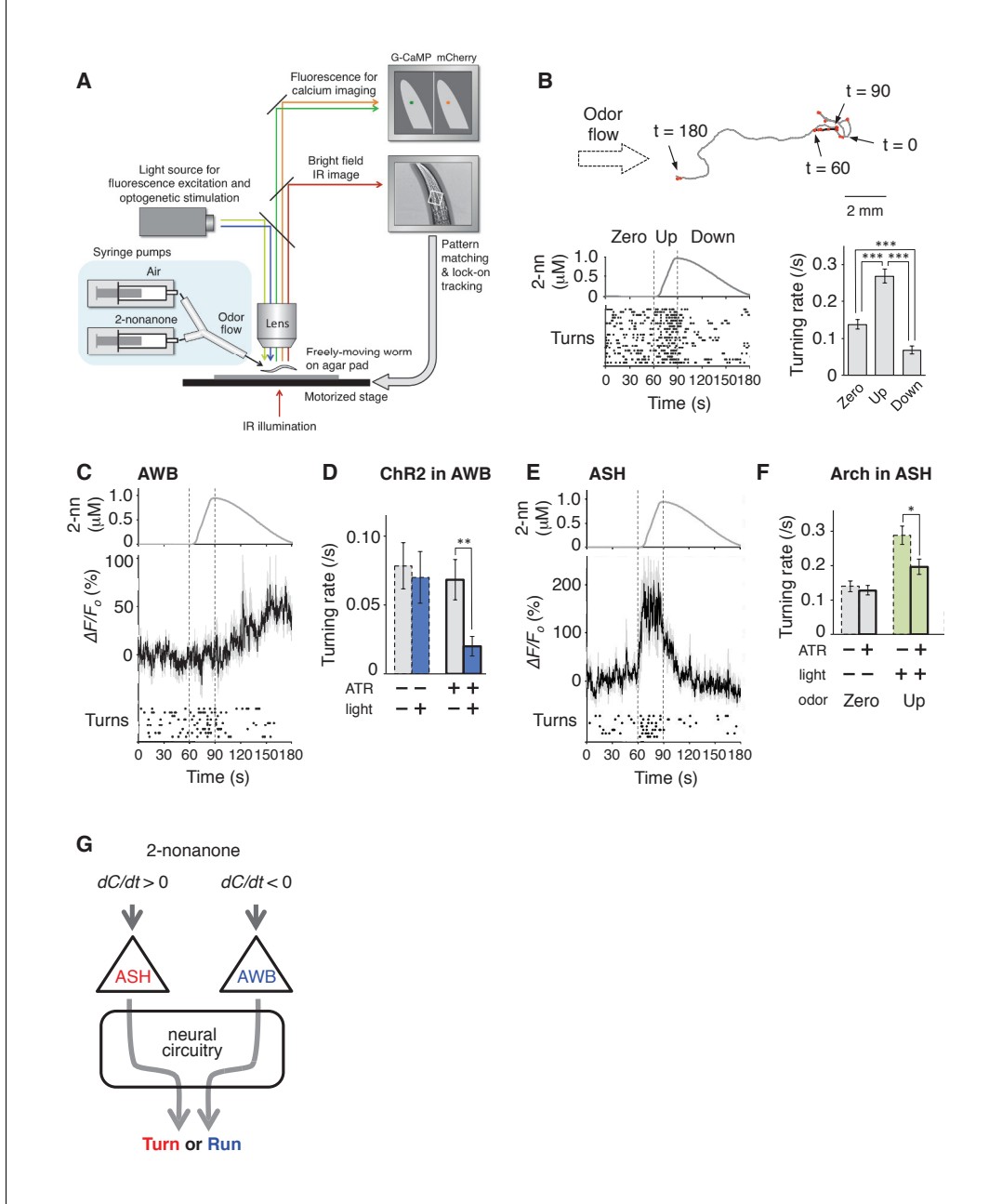

**Figure 3.** AWB and ASH sensory neuron pairs regulate turning rate in response to *dC/dt* of 2-nonanone. (**A**) Schematic drawing of the OSB2 system. (**B**) Behavioral response to temporal changes in the 2-nonanone concentration. (Top) Track of a wild-type animal. The first 60 s (gray) is a period of no odor ('odor-zero phase' in the bottom left panel), 60–90 s (black) is a period with a constant increase in odor concentration from 0 to 1 µM ('odor-up phase'), and 90–180 s (gray) is a period with a constant decrease in odor concentration from 1 to 0 µM ('odor-down phase'). Red dots indicate turns. (Bottom left) Rastergram of turns. The upper portion of the panel shows the measured 2-nonanone concentration in the flow. In the lower portion of the panel, each turn is denoted by a dot, and each row represents the behavioral record of a single animal during the 180 s of analysis. The results of 20 animals are shown. (Bottom right) Ensemble averages ± SEM for the turning rate (turns per second) during each phase in the left panel (n = 20). The turning rates in the three conditions differed significantly from each other (***p<0.001, Kruskal-Wallis test with *post hoc* Steel-Dwass test). (**C**) Response of AWB neurons. The averages ± SEM of $\Delta F/F_0$ for GCaMP3 (middle) and rastergram of turns of the animals (bottom) are shown (n = 9). (**D**) Effect of optogenetic activation of AWB neurons on the turning rate in the absence of odor. Transgenic animals expressing ChR2(C128S), a bi-stable variant of ChR2, in AWB were cultivated in the absence or presence of all-*trans*-retinal (ATR) (dashed or solid bars, respectively); exogenous ATR is required for functional ChR in *C. elegans*

*Figure 3 continued on next page*

*Figure 3 continued*

(***Nagel et al., 2005***). Average turning rates ± SEM of the 30 s periods before (gray bars) or after blue light illumination for 3 s (blue bars) are shown (n = 20 each). **p<0.01 (Mann-Whitney test). (**E**) Calcium imaging of ASH neurons using GCaMP3 (n = 7). (**F**) Effect of optogenetic silencing of ASH neurons on the turning rate. The transgenic animals expressing Arch in ASH neurons were cultivated in the absence or presence of ATR (dashed or solid bars, respectively) and illuminated with green light during the up-phase (n = 16 each). The odor pattern was the same as that in panels B, C, and E. *p<0.05 (Mann-Whitney test). (**G**) Model of the regulation of odor avoidance by ASH and AWB neurons. All the statistical details are shown in ***Supplementary file 1***. The following figure supplement is available for ***Figure 3***.

The following figure supplement is available for figure 3:

**Figure supplement 1.** Spatial arrangement of the odor stimulation and behavioral response in the OSB2 system.

## Two sensory pathways mediate odor increases or decreases on a virtual odor gradient

To determine whether sensory stimuli are the causal reason for the behavioral response during odor avoidance, and the neural mechanisms linking stimulus and behavior, we developed a novel integrated microscope system for the quantitative optophysiological analyses of freely moving *C. elegans* on virtual odor gradients. We integrated the OSB system, an auto-tracking microscope system for calcium imaging and optogenetic analyses of neuronal activity in *C. elegans* (***Video 2***), with an odor-delivery sub-system (***Busch et al., 2012***; ***Tanimoto et al., 2016***). Using this new OSB2 system, a moving *C. elegans* was continuously exposed to an odor flowing from syringe pumps (***Figure 3A*** and ***Figure 3—figure supplement 1A***, and ***Video 3***). The odor concentration in the flow changed according to a predefined program, based on estimated values experienced in the plate assay paradigm (***Figure 2***). Using the OSB2 system, we first tested whether the $dC/dt$ of the odor itself could regulate an animal's behavior. When animals experienced temporal increases in the odor concentration, they exhibited more frequent turns like in pirouettes (***Figure 3B***). Conversely, when they

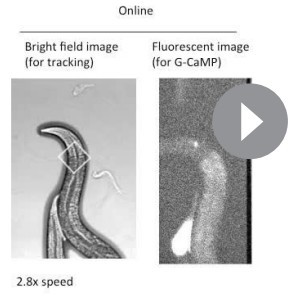

**Video 2.** A demonstration video for calcium imaging with the OSB2 system.    (Left) The bright field images for the tracking and the fluorescence images for calcium imaging were acquired simultaneously but separately in the tracking and calcium imaging subsystems, respectively. The video was compiled separately because of the different frame rates and ran on a Power Point file. Because of a technical limitation on the video playing, the videos were not completely synchronized. (Right) From the fluorescence images, the cell body of AWB neuron was tracked and centered off-line. The video speed is 2.8x.

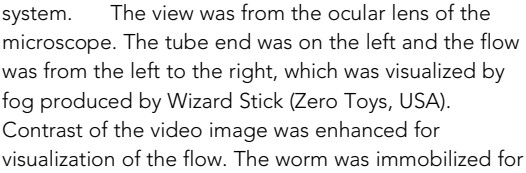

**Video 3.** Visualization of the odor flow on the OSB2 system.    The view was from the ocular lens of the microscope. The tube end was on the left and the flow was from the left to the right, which was visualized by fog produced by Wizard Stick (Zero Toys, USA). Contrast of the video image was enhanced for visualization of the flow. The worm was immobilized for reference. The video speed is 1x.

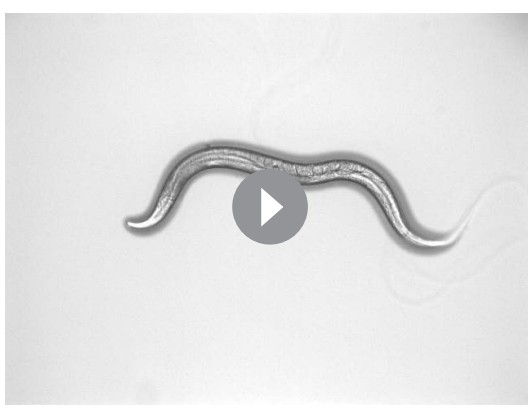

**Video 4.** Optogenetic activation of AWB neurons. After 60 s without any stimulus, a transgenic animal expressing the bistable variant of channelrhodopsin, ChR2(C128S), was illuminated with blue light for 3 s to cause sustained AWB activation, and the behavior was monitored for the following 30 s. The video speed is 8x.

experienced temporal decreases, they suppressed turns and exhibited long migrations similar to runs (*Figure 3B*). During the odor-down phase, no statistical bias was detected in migratory direction (*Figure 3—figure supplement 1B*). These results indicate that *C. elegans* responds to temporal changes in the odor concentration and regulates turning rates.

We then performed calcium imaging of neural activity and revealed that two pairs of sensory neurons regulate the avoidance behavior by responding to either increases or decreases in the odor concentration. In *C. elegans'* sensory neurons, dynamic changes in $[Ca^{2+}]_i$ in the cell bodies are generally similar to those in axons, which cause neurotransmitter release from the neurons (*Kato et al., 2014*; *Zahratka et al., 2015*). First, we investigated a pair of AWB olfactory neurons. AWB neurons are known to be primarily responsible for 2-nonanone avoidance and to exhibit an odor-OFF response when animals were stimulated with 2-nonanone-saturated buffer in a stepwise manner (*Ha et al., 2010*; *Troemel et al., 1997*). However, it was not clear whether and how AWB neurons respond to gradual and/or subtle changes in odor concentration in the air phase. We found that AWB neurons were gradually and continuously activated during the odor-down phase for 90 s (*Figures 3C*, 90–180 s). Optogenetic activation of AWB neurons by the bistable variant of the light-gated cation channel ChR2(C128S) (*Berndt et al., 2009*) significantly suppressed turns in the absence of the odor (*Figure 3D* and *Video 4*). In addition, we also found that ASH neurons, a pair of nociceptive neurons (*Bargmann, 2006*; *Kaplan, 1996*), were activated by an odor increase (*Figures 3D*, 60–90 s). Consistently, optogenetic inactivation of ASH neurons during the odor-up phase by the light-driven H[+] pump Arch (*Chow et al., 2010*) significantly suppressed the increase in the turning rate (*Figure 3F*). Taken together, these results indicate that two distinct sensory pathways respond to the physiological range of odor concentration changes and opposingly regulate the avoidance behavior (*Figure 3G*).

## Temporal differentiation and integration of odor information regulate avoidance behavior

Through mathematical analyses of ASH and AWB responses, we found, unexpectedly, that the two neuron pairs decode the temporal dynamics of odor concentration information using different computations. When *C. elegans* was stimulated with different rates of positive $dC/dt$, the ASH neurons were always activated rapidly (*Figure 4A*, black lines in middle panels). These responses peaked soon after the onset of the odor-up phase and were mostly maintained during this phase, which can be approximated by time-differentials of the odor concentrations (*i.e.*, $dC/dt$: *Figure 4A*, red lines in middle panels; see also *Table 1*). The time-differential response of sensory neurons is consistent with recent studies (*Lockery, 2011*; *Schulze et al., 2015*) and the general idea that sensory neurons respond in a phasic manner by detecting stimulus changes (*Delcomyn, 1998*). A portion of ASH response (*e.g., decay kinetics during the odor-plateau phase*) was not well fitted by the time-differential equation, which reflected the fact that ASH response is regulated by multiple mechanisms (see below).

The behavioral response also changed rapidly: At the onset of the odor-up phase, the turning rates exceeded the upper limit of the 99% prediction interval for the rate during the odor-zero phase (*Figure 4A*, red vertical dotted lines in lower panels): The 99% prediction interval is a value in which future data (odor-up phase in this case) will fall with 99% probability based on the observed data (odor-zero phase) (*Montgomery and Runger, 2002*). In the 45 s and 90 s odor-up conditions (left-most and middle left panels), the turning rates were significantly different from the no odor

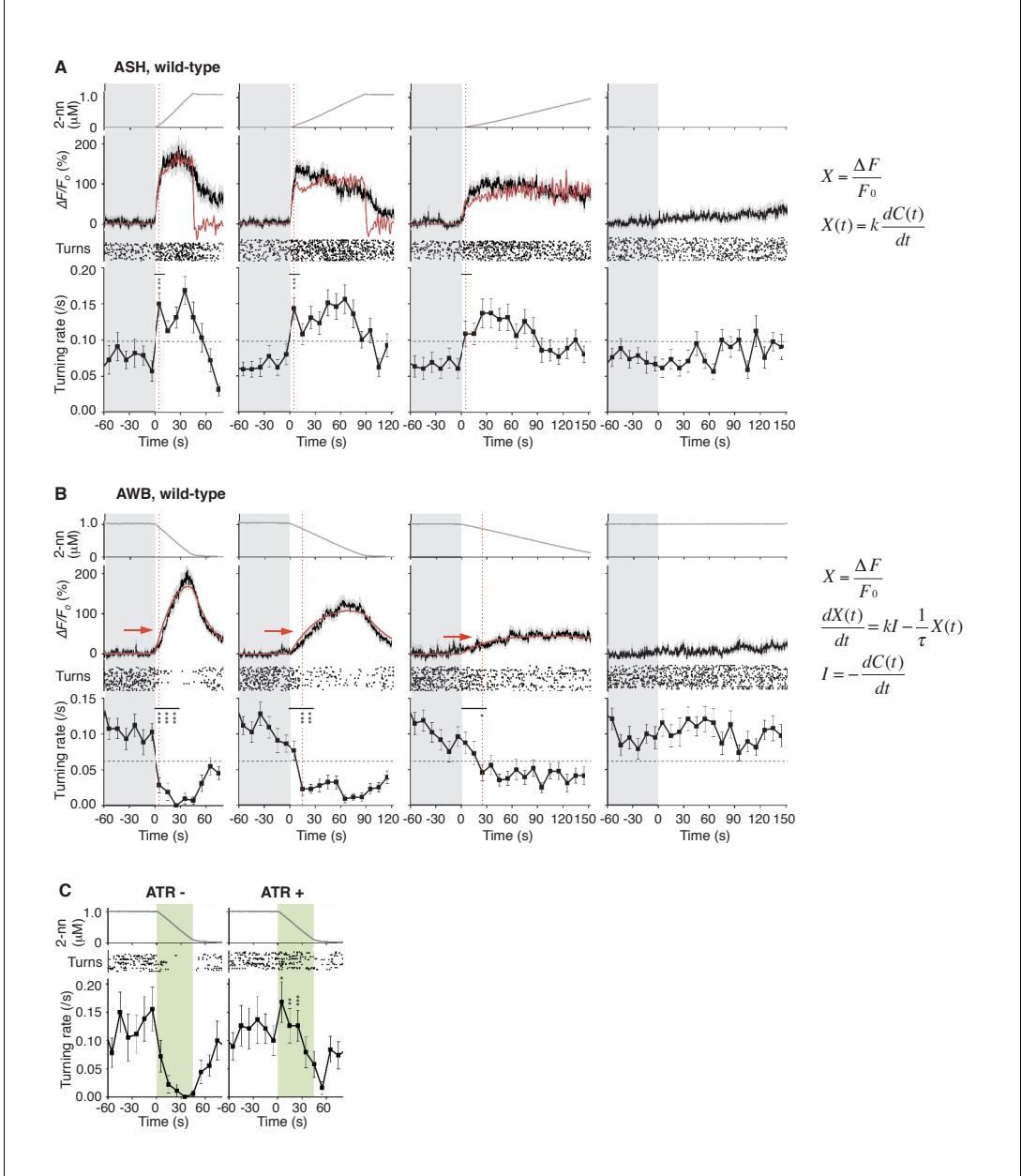

**Figure 4.** ASH neurons are activated according to *dC/dt* for initiating turns, and AWB neurons are activated according to the leaky integration of the negative *dC/dt* for suppressing turns with a *dC/dt*-dependent delay. (**A**) ASH responses (middle panels) and turns (lower panels) in response to odor concentration increases from 0 to 1 μM in 45 s (left most; n = 32), 90 s (middle left; n = 39), 180 s (middle right; n = 35) or no odor increase ('no odor control'; right most; n = 41) are shown. (Middle panels) In the three conditions with odor increases, the response patterns of ASH neurons (the average ± SEM: black lines and gray shadows, respectively) were fitted by time-differentials of the average of measured odor concentration (red lines), calculated by the rightmost equations. (Lower panels) Ensemble averages of the turning rates ± SEM in each 10 s bin were calculated. The original data is shown in the raster plot. Black horizontal dashed lines in lower panels indicate the upper limit of 99% prediction interval of all the turning rates during the odor-zero phase (t = −60 ~ 0; gray area). Red vertical dotted lines indicate the time when each turning rate first exceeded the upper limit of prediction interval. In the first bin of odor-up phase (indicated by a black horizontal bar in the lower panels), the turning rate in the 45 s and 90 s conditions increased significantly compared to the no-odor control (***p<0.001, Kruskal-Wallis test with *post hoc* Steel-Dwass test). (**B**) AWB responses and turns in response to odor concentration decreases from 1 μM (odor-plateau phase; gray area) to 0 μM (odor-zero phase) in 45 s (left-most; n = 42), in 90 s (middle left; n = 43), in 180 s (middle right; n = 48) and no odor decrease ('odor-plateau control'; right-most; n = 38). AWB responses to different *dC/dt* rates were fitted by a leaky integrator equation of negative *dC/dt* (red lines). Black horizontal dashed lines in the lower panels indicate the lower limit of the 99% prediction interval of the turning rates during the odor-plateau phase. The times when the turning rates became lower than the limit (red vertical dotted lines) were delayed when the *dC/dt* rate was smaller (*p<0.005 and ***p<0.001, Kruskal-Wallis test with *post hoc* Steel-Dwass test). The statistical test was performed in the first 3 bins of the odor-down phase (a black horizontal bar in lower panels). (**C**), Effect of optogenetic silencing

*Figure 4 continued on next page*

*Figure 4 continued*

of the AWB response on negative *dC/dt*. Transgenic animals expressing Arch in AWB neurons were cultivated in the absence (n = 18) or presence (n = 19) of ATR and illuminated with green light during the odor-down phase. *p<0.05, **p<0.01, and ***p<0.001 (Mann-Whitney test). In the panels, the gray shading means the period with no odor change, in which the prediction intervals were calculated. All the statistical details are shown in *Supplementary file 1*. The following figure supplements are available for *Figure 4*.

The following figure supplements are available for figure 4:

**Figure supplement 1.** AWB responses were not fitted sufficiently by time-differential equations.

**Figure supplement 2.** Estimated intracellular calcium concentrations in AWB neurons calculated from measured $\Delta F/F_0$ in *Figure 4B*.

**Figure supplement 3.** ASH responses were partially fitted by the time-integral equations.

control (see *Supplementary file 1* for statistical details). It should be noted that, even with the slowest odor concentration increase (middle right panel), at which the overall turning rate was around the threshold, the turning rate increased and surpassed the threshold at the onset of the odor-up phase although no statistical difference was detected. These results suggest that the nociceptive ASH neurons compute the time-differential of odor concentration to rapidly cause an aversive response based on a small change in odor concentration.

In contrast, the AWB responses to decreases in odor concentration were time-integral. The responses gradually increased and peaked with a considerable delay, which depended on the *dC/dt* rates after the onset of the odor-down phase (*Figure 4B*, black lines in middle panels). These responses could not be fitted by the *dC/dt* itself but by a leaky integrator equation, in which *-dC/dt* acted as the input (*Figure 4B*, red lines in middle panels and *Figure 4—figure supplement 1*; see also *Table 2* for parameters and *Table 3* for goodness of fit). In the leaky integrator model, the neuronal response is given by the sum of the past inputs, which decreases exponentially due to leakage. Furthermore, we estimated $[Ca^{2+}]_i$ by including a sigmoidal relationship between calcium concentration and fluorescence intensity into the model (*Akerboom et al., 2012*; *Kato et al., 2014*). The estimated calcium concentration exhibited a similar pattern to the measured fluorescence change with an estimated basal concentration of about 60–90 nM (*Figure 4—figure supplement 2* and *Table 2*), similar to that generally reported for resting neurons in general (~100 nM) (*Clapham, 2007*). Taken together, these results suggest that AWB neurons compute the leaky integration of the negative *dC/dt* to temporally integrate the sensory information as the accumulation of $[Ca^{2+}]_i$.

**Table 1.** Models and parameters used in the fitting of ASH responses with time-differential equations.

| Conditions | ASH, wild-type | | | ASH, *odr-3(n2150)* | ASH, *odr-3(n1605)* | ASH, wild-type slow component |
|---|---|---|---|---|---|---|
| Durations of up/down phase | Up 45 s | Up 90 s | Up 180 s | Up 90 s | Up 90 s | Up 90 s |
| Model | $X(t) = kI$ | $X(t) = kI$ | $X(t) = kI$ | $X(t) = kI$ | $X(t) = kI$ | $\frac{dX(t)}{dt} = kI - \frac{1}{\tau}X(t)$ |
| | $I = \frac{dC(t)}{dt}$ | $I = \frac{dC(t)}{dt}$ | $I = \frac{dC(t)}{dt}$ | $I = \frac{dC(t)}{dt}$ | $I = \frac{dC(t)}{dt}$ | $I = \frac{dC(t)}{dt}$ |
| Parameters used for fitting to $\Delta F/F_0$ | | | | | | |
| $k$ | 56.1 [μM⁻¹·s] | 80.9 [μM⁻¹·s] | 121.9 [μM⁻¹·s] | 49.1 [μM⁻¹·s] | 46.7 [μM⁻¹·s] | 2.96 [μM⁻¹] |
| $\tau$ | - | - | - | - | - | 10.4 [s] |
| Parameters used for fitting to estimated calcium concentration | | | | | | |
| $k$ | 3.9 [s] | 6.2 [s] | 7.9 [s] | 2.8 [s] | 2.9 [s] | - |
| $\tau$ | - | - | - | - | - | - |
| $f_{max}$ | 9.4 | 9.5 | 9.2 | 9.1 | 8.7 | - |
| $f_{min}$ | 0.79 | 0.79 | 0.77 | 0.76 | 0.73 | - |
| $X_{base}$ | 103.8 [nM] | 92.2 [nM] | 107.0 [nM] | 108.2 [nM] | 109.7 [nM] | - |

Table 2. Models and parameters used in the fitting of AWB responses with leaky integrator equations.

| Conditions | AWB, wild-type | | | AWB, unc-13 (e51) | AWB, unc-31 (e928) | AWB, odr-3 (n2150) | AWB, odr-3 (n1605) |
|---|---|---|---|---|---|---|---|
| **Durations of up/down phase** | Down 45 s | Down 90 s | Down 180 s | Down 90 s | Down 90 s | Down 90 s | Down 90 s |
| Model | $\frac{dX(t)}{dt} = kI - \frac{1}{\tau}X(t)$ | $\frac{dX(t)}{dt} = kI - \frac{1}{\tau}X(t)$ | $\frac{dX(t)}{dt} = kI - \frac{1}{\tau}X(t)$ | $\frac{dX(t)}{dt} = kI - \frac{1}{\tau}X(t)$ | $\frac{dX(t)}{dt} = kI - \frac{1}{\tau}X(t)$ | $\frac{dX(t)}{dt} = kI - \frac{1}{\tau}X(t)$ | $\frac{dX(t)}{dt} = kI - \frac{1}{\tau}X(t)$ |
| | $I = -\frac{dC(t)}{dt}$ | $I = -\frac{dC(t)}{dt}$ | $I = -\frac{dC(t)}{dt}$ | $I = -\frac{dC(t)}{dt}$ | $I = -\frac{dC(t)}{dt}$ | $I = -\frac{C(t)-C(t-\Delta t)}{\Delta t}$ | $I = -\frac{C(t)-C(t-\Delta t)}{\Delta t}$ |
| | | | | | | $(\Delta t = 67\ s)$ | $(\Delta t = 66\ s)$ |
| **Parameters used for fitting to $\Delta F/F_0$** | | | | | | | |
| $k$ | 4.6 [μM$^{-1}$] | 3.6 [μM$^{-1}$] | 3.0 [μM$^{-1}$] | 2.6 [μM$^{-1}$] | 4.9 [μM$^{-1}$] | 8.7 [μM$^{-1}$] | 7.7 [μM$^{-1}$] |
| $\tau$ | 19.7 [s] | 28.1 [s] | 25.0 [s] | 34.5 [s] | 28.1 [s] | 25.2 [s] | 23.5 [s] |
| **Parameters used for fitting to estimated calcium concentration** | | | | | | | |
| $k$ | 0.40 | 0.37 | 0.33 | 0.26 | 0.44 | 1.00 | 0.87 |
| $\tau$ | 21.1 [s] | 28.5 [s] | 25.0 [s] | 37.3 [s] | 30.7 [s] | 17.7 [s] | 17.7 [s] |
| $f_{max}$ | 9.7 | 10.1 | 8.5 | 9.6 | 8.9 | 10.9 | 10.4 |
| $f_{min}$ | 0.81 | 0.84 | 0.71 | 0.80 | 0.74 | 0.91 | 0.87 |
| $X_{base}$ | 66.1 [nM] | 56.7 [nM] | 89.6 [nM] | 66.7 [nM] | 81.2 [nM] | 41.8 [nM] | 51.5 [nM] |

The *dC/dt* rate-dependent delay in AWB activation was correlated with the behavioral responses. The time when the turning rate became lower than the 99% prediction interval of the odor-plateau phase was delayed according to the *dC/dt* rate, which was associated with statistical differences from the odor-plateau control (*Figure 4B*, red vertical dotted lines in lower panels). This result suggests that the turning rate is suppressed when the AWB activity exceeds a certain value (*Figure 4B*, red horizontal arrows in middle panels). Interestingly, even when animals were stimulated with the most shallow odor gradient in which the average turning rate appeared gradually decreased (middle right panels in *Figure 4B*), turn intervals of each animal could be classified into two groups like pirouettes and runs (compare *Figure 1—figure supplement 1E* with 1A). This result suggests that, even with subtle changes in odor concentration, the behavioral response of *C. elegans* does not change gradually, but instead transits between high- and low-turning states. Optogenetic suppression of the AWB neurons with Arch during the odor-down phase significantly affected the transition (*Figure 4C*). Taken together, these results indicate that AWB neurons regulate the transition from a pirouette to a run based on temporal integration of the negative *dC/dt*.

## Computer simulations reveal the behavioral effects of the temporal integration of sensory information

To further understand whether the neural computations of odor concentration contribute to the choice of migratory direction during decision-making, we performed computer simulations of the

Table 3. Parameters and goodness of fit results for mathematical models of ASH and AWB responses.

| Conditions | ASH, wild-type | | | AWB, wild-type | | |
|---|---|---|---|---|---|---|
| **Durations of up/down phase** | Up 45 s | Up 90 s | Up 180 s | Down 45 s | Down 90 s | Down 180 s |
| Number of samples (frames) used for calculation of BIC | N = 135 (t = −60 ~ 75 s) | N = 180 (t = −60 ~ 120 s) | N = 270 (t = −60 ~ 210 s) | N = 135 (t = −60 ~ 75 s) | N = 180 (t = −60 ~ 120 s) | N = 270 (t = −60 ~ 210 s) |
| $X(t) = kI$<br>$I = \frac{dC(t)}{dt}$ | k = 56.1 [μM$^{-1}$·s]<br>BIC = −222.2 | k = 80.9 [μM$^{-1}$·s]<br>BIC = −446.5 | k = 121.9 [μM$^{-1}$·s]<br>BIC = −637.8 | k = −58.4 [μM$^{-1}$·s]<br>BIC = −170.6 | k = −73.4 [μM$^{-1}$·s]<br>BIC = −372.5 | k = −67.3 [μM$^{-1}$·s]<br>BIC = −1157 |
| $\frac{dX(t)}{dt} = kI - \frac{1}{\tau}X(t)$<br>$I = \frac{dC(t)}{dt}$ | k = 5.8 [μM$^{-1}$]<br>τ = 12.0 [s]<br>BIC = −331.1 | k = 18.9 [μM$^{-1}$]<br>τ = 4.4 [s]<br>BIC = −472.8 | k = 11.5 [μM$^{-1}$]<br>τ = 11.7 [s]<br>BIC = −804.9 | k = −4.56 [μM$^{-1}$]<br>τ = 19.7 [s]<br>BIC = −591.9 | k = −3.58 [μM$^{-1}$]<br>τ = 28.1 [s]<br>BIC = −806.2 | k = −3.01 [μM$^{-1}$]<br>τ = 25.0 [s]<br>BIC = −1458 |

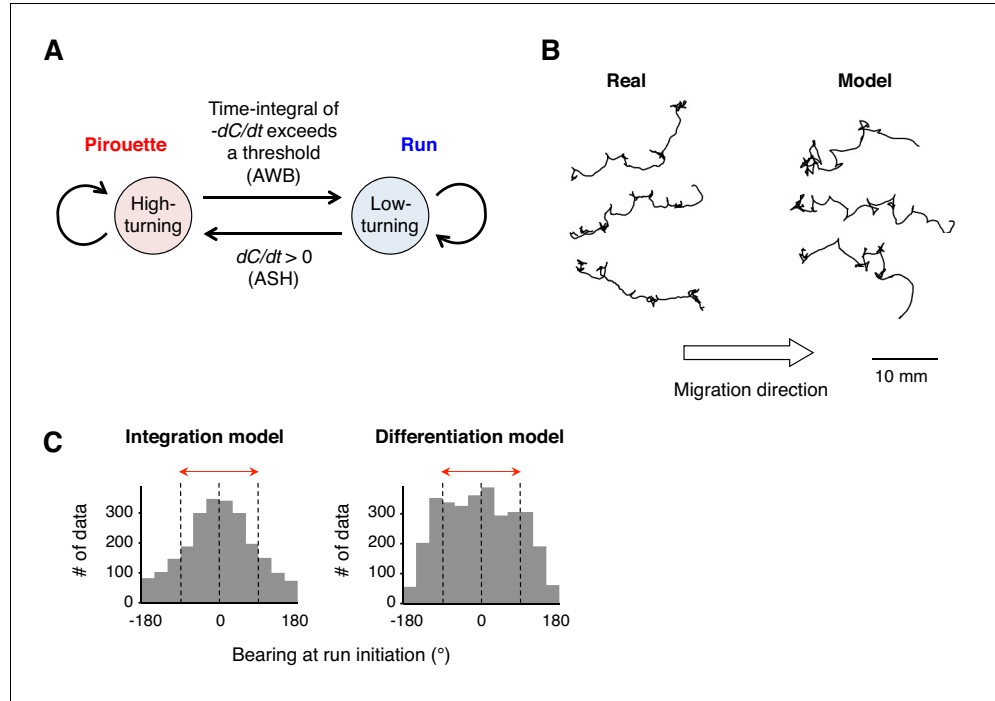

**Figure 5.** A computer model reproduced the directional choice in the odor avoidance task in a temporal integration-dependent manner. (A) Model of the behavioral transition in 2-nonanone avoidance. During a pirouette, a model animal frequently repeated turns and short migrations. When a model animal initiated a migration away from the odor source and sensed $dC/dt$ <0, the high-turning state transited to a low-turning state (*i.e.*, a run) when the leaky integrator equation exceeded a threshold. During a run, the animals turned with much lower frequency than in pirouette and at a probability related to $dC/dt$ >0 due to straying away from the original direction. (B) Three typical tracks of real (left) and model (right) animals. As shown in *Figure 1A*, the odor source is on the left side. (C) Histograms of the initial bearings of runs in the model animals. The high-turning-to-low-turning transition was dependent on temporal integration (left) or on differentiation (right). p<0.001 (Mardia-Watson-Wheeler test). All the statistical details are shown in *Supplementary file 1*.

odor avoidance behavior (*Figure 5A*). During pirouettes, the model animal frequently repeated turns and short migrations in random directions, and a pirouette transited to a run when the time-integral of $-dC_{worm}/dt$ over time reached a threshold value. The model produced similar migration patterns to the odor avoidance behavior seen in real animals, in which most of the runs were down the odor gradient (*Figure 5B*). Interestingly, when we changed the model animal's computation for transition of a pirouette to a run from the temporal integration of $dC/dt$ (the integration model) to the simple $dC/dt$ itself (the differentiation model), the directional choice at run initiation significantly worsened (*Figure 5C*; p<0.001, Mardia-Watson-Wheeler test). This is because, in the differentiation model, the animals initiated runs even when they transiently sensed $dC/dt$ <0, due to the periodical head swing for example (*Iino and Yoshida, 2009*; *Yamazoe-Umemoto et al., 2015*). In contrast, animals in the leaky integration model initiated runs only when they sensed $dC/dt$ <0 for a certain period of time, resulting in an appropriate directional choice. In reality, the movement of the animal's anterior end (where the sensory endings of the ASH and AWB neurons are exposed to the environment) is more random than that in our model (*Videos 2* and *4*), suggesting that the temporally integrating property of AWB neurons may play a significant role in robust directional choice based on a noisy sensory input.

## Temporal integration of sensory information occurs in AWB neurons

Temporally integrating neural activity for decision-making and working memory is considered to be based on recurrent synaptic circuits in vertebrates (*Aksay et al., 2007*; *Wang, 2008*). To determine whether a synaptic circuit input is required for the temporally integrating property of AWB neurons,

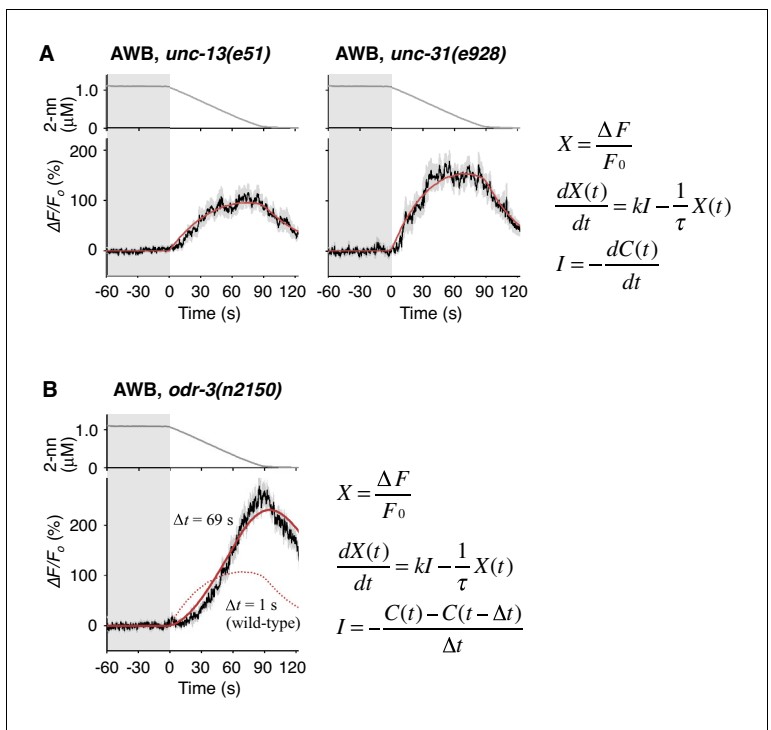

**Figure 6.** Cell-autonomous computations in AWB neurons. (**A**) The AWB responses of *unc-13* (left) and *unc-31* (right) mutants to the odor decreases, which are the same as those shown in the middle left panel of ***Figure 4B***, were essentially similar to those of wild-type animals and fitted by the leaky integrator equations (n = 31 and 17, respectively). (**B**) The AWB responses of *odr-3(n2150)* mutants (n = 24) were fitted by the leaky integrator equation (red solid line) with the time interval (Δt) being much longer than that of wild-type AWB (red dotted line). The parameters are described in ***Table 2***. In the experiments for panels A and B, the behavioral responses were not analyzed because the *unc-13*, *unc-31*, and *odr-3* mutations affect the activities of multiple neurons, including ASH and AWB.

The following figure supplement is available for figure 6:

**Figure supplement 1.** Responses of AWB and ASH neurons in *odr-3* mutants.

we abolished the transmission of synaptic and/or dense core vesicles in most, if not all, neurons via a mutation in *unc-13* or *unc-31*, the orthologs of mammalian Munc-13 and CAPS (calcium activated protein for secretion), respectively (***Richmond, 2005***). In the *unc-13* and *unc-31* mutants, however, the AWB responses were essentially similar to that in wild-type animals and fitted with the same leaky integrator equations (***Figure 6A*** and ***Table 2***), indicating that the temporally integrating neural activity occurs within the AWB neuron itself and does not require a recurrent circuit.

## Molecular mechanisms of neural computation in the sensory neurons

To reveal the molecular mechanisms of the temporal integration of sensory information in AWB neurons, we performed genetic manipulations. First, we identified the gene required for the calculation of *dC/dt.* Previous genetic studies suggest that the signaling pathway mediating odor response in AWB neurons is similar to that in mammalian olfactory/photoreceptor neurons. Upon binding to an odorant, a G-protein coupled receptor activates Gα protein, which eventually leads to the opening of cyclic nucleotide-gated cation channels for depolarization (***Bargmann, 2006***). ODR-3, one of the 20 Gα homologs in *C. elegans*, is expressed in AWB, ASH and a few other pairs of sensory neurons. It is localized at the sensory ending, and its dysfunction causes severe chemotaxis defects, suggesting that ODR-3 transduces the sensory signal from odorant receptor (***Roayaie et al., 1998***). Two alleles of loss-of-function mutations of the *odr-3* gene, however, caused gradual activations of AWB neurons, whose peaks were even larger than those of wild-type animals (***Figure 6B*** and ***Figure 6—***

**Table 4.** Parameters and goodness of fit for mathematical models of AWB responses in *odr-3* mutants.

| Conditions | AWB, *odr-3(n2150)* | AWB, *odr-3(n1605)* |
|---|---|---|
| Durations of up/down phase | down 90 s | down 90 s |
| Number of samples (frames) used for calculation of BIC | N = 180 (t = −60 ~ 120 s) | N = 180 (t = −60 ~ 120 s) |
| $X(t) = kI$ <br> $I = -\frac{dC(t)}{dt}$ | k = 89.1 [μM$^{-1}$·s] <br> BIC = 25.1 | k = 77.3 [μM$^{-1}$·s] <br> BIC = −39.2 |
| $\frac{dX(t)}{dt} = kI - \frac{1}{\tau}X(t)$ <br> $I = -\frac{dC(t)}{dt}$ | k = 2.08 [μM$^{-1}$] <br> $\tau \to \infty$ <br> BIC = −477.7 | k = 1.77 [μM$^{-1}$] <br> $\tau \to \infty$ <br> BIC = −578.5 |
| $\frac{dX(t)}{dt} = kI - \frac{1}{\tau}X(t)$ <br> $I = -\frac{C(t)-C(t-\Delta t)}{\Delta t}$ | k = 8.71 [μM$^{-1}$] <br> $\tau$ = 25.2 [s] <br> $\Delta t$ = 67 [s] <br> BIC = −645.6 | k = 7.73 [μM$^{-1}$] <br> $\tau$ = 23.5 [s] <br> $\Delta t$ = 66 [s] <br> BIC = −779.9 |

*figure supplement 1A*, black lines), suggesting that ODR-3 may play an inhibitory role in sensory signaling. Interestingly, while the AWB response patterns in wild-type animals were fitted by a leaky integrator equation with a time interval for the input ($\Delta t$)=1 s (solid red lines in *Figure 4B* and dotted red line in 6B), AWB response patterns in *odr-3* mutants were better fitted by an equation with $\Delta t$ longer than 1 min than by $\Delta t$ = 1 s (solid red line in *Figure 6B*, *Figure 6—figure supplement 1A*; see also *Table 2* for parameters and *Table 4* for goodness of fit). Thus, the time-differential property for the sensory input was greatly degraded with the *odr-3* mutations while the time-integral property was not affected. This result suggests that the time-differential computation of odor concentration depends mostly on the ODR-3 Gα protein (see Discussion for details). The result also supports the idea that AWB neurons possess a temporally integrating property for sensory inputs. The overall response of ASH neurons, which also express ODR-3, was partially affected by the *odr-3* mutations (*Figure 6—figure supplement 1B*), suggesting that ODR-3 may play a primary role in sensory signaling in ASH neurons.

Furthermore, we genetically identified the genes responsible for the time-integral calcium accumulation. In general, [Ca$^{2+}$]$_i$ increases in neurons depend on its influx through the plasma membrane via VGCCs, which can then trigger rapid calcium release from the endoplasmic reticulum (ER) via IP$_3$ receptors (IP$_3$Rs) and/or ryanodine receptors (RyRs) as calcium-induced calcium release (CICR) (*Catterall, 2011*; *Clapham, 2007*). The pore-forming α$_1$ subunits of VGCCs are classified into L-type, N/P/Q-type, and T-type subgroups. Previous studies have demonstrated the requirement for EGL-19 and/or UNC-2, the sole orthologs of L- and N-type channels, respectively, in the responses of sensory neurons of *C. elegans* (*Busch et al., 2012*; *Frøkjaer-Jensen et al., 2006*; *Hilliard et al., 2005*; *Kato et al., 2014*; *Larsch et al., 2015*; *Suzuki et al., 2003*; *Zahratka et al., 2015*) although their precise roles in intracellular calcium dynamics have remained unclear. We found that an *egl-19* reduction-of-function mutation (rf) (*Lee et al., 1997*) significantly affected the AWB response (*Figure 7A*). Consistently, treatment with the EGL-19 antagonist Nemadipine-A (NemA) (*Kwok et al., 2006*) exhibited a similar, or possibly even stronger, effect. In contrast, mutations in *unc-2*, *itr-1* (the IP$_3$R ortholog) or *unc-68* (the RyR ortholog) (*Dal Santo et al., 1999*; *Sakube et al., 1997*; *Schafer and Kenyon, 1995*) did not significantly affect the response (*Figure 7A and C* left panel). These results suggest that the calcium accumulation in AWB neurons mostly depends on influx through the EGL-19 L-type VGCCs, but not other calcium channels.

In ASH neurons, whose response pattern was also not affected by synaptic connections (*Figure 7—figure supplement 1*), the suppression of EGL-19 with NemA mainly affected the later phase of the response, but did not significantly affect the initial response (*Figure 7B and C* right panel). The mathematical addition of the AWB-like time-integral response to the NemA-treated ASH response resembled the wild-type ASH response (*Figure 7D*), suggesting that the ASH response consists of both fast and slow components. Because increases in the turning rate occurred at the onset of the odor-up phase (*Figure 4A*), the fast component may have a major influence on the behavioral response. This fast component was not affected by a loss-of-function mutation in *unc-2* or

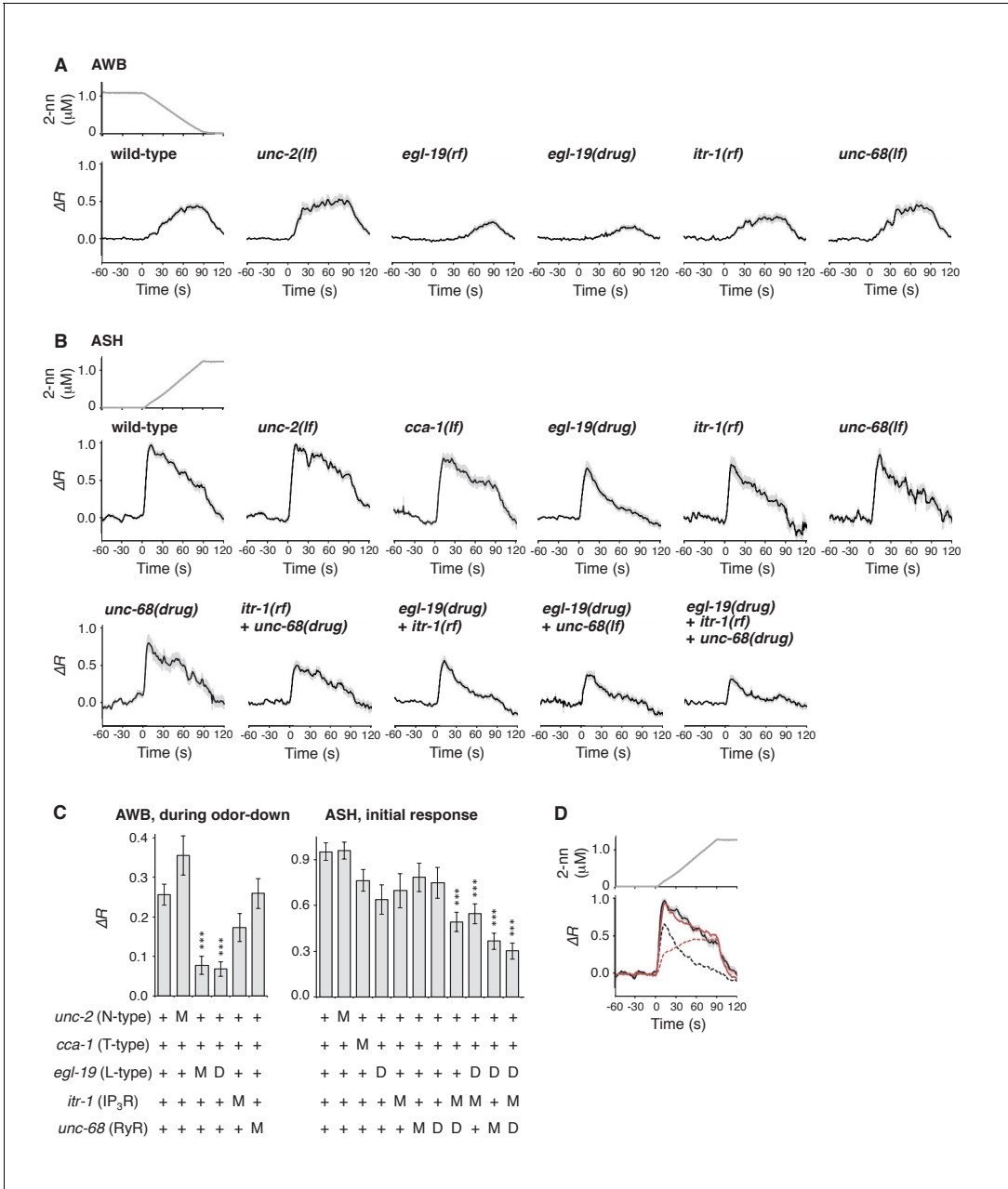

**Figure 7.** Calcium channels are involved in the dynamic regulation of [Ca$^{2+}$]$_i$ in a cell type-dependent manner. (**A** and **B**) Responses of AWB (panel A) or ASH (panel B) neurons in strains with genetic and/or pharmacological suppression of N/P/Q-type VGCC UNC-2, T-type VGCC CCA-1, L-type VGCC EGL-19, IP$_3$R ITR-1, and RyR UNC-68. *lf* is a loss-of-function mutation, and *rf* is a reduction-of-function mutation. Note that loss-of-function mutations of *egl-19* and *itr-1* are not available due to possible lethality. (**C**) Suppression of the AWB response during the odor-down phase (left) and of the initial ASH response (10–15 s of the odor-up phase) (right) with the genetic mutation (M) and/or the drug treatment (D). ***p<0.001 (Kruskal-Wallis test with *post hoc* Steel-Dwass test). (**D**) *egl-19* is also responsible for the slow time-integral component in ASH neurons. Addition of a putative time-integral response using the leaky integrator equation to the transient response in the '*egl-19(drug)*' reproduced the ASH response of the naive wild-type animals. Black line: wild-type response shown in panel B; black dashed line: *egl-19(drug)* response in panel B; red dashed line: the time-integral model of positive *dC/dt*; red line: sum of the black and red dashed lines. The parameters are described in **Table 1**. All the statistical details are shown in **Supplementary file 1**.

The following figure supplement is available for figure 7:

**Figure supplement 1.** ASH response does not depend on synaptic transmission.

in a T-type VGCC homolog, *cca-1* (*Steger et al., 2005*), suggesting that this calcium response is not mediated by the typical VGCCs. Suppressing CICR by mutations of *itr-1* or *unc-68*, or using dantorolene, a RyR antagonist (*Krause et al., 2004*), partially affected the magnitude, but not the temporal pattern, of the ASH response regardless of NemA treatment (*Figure 7B and C* right panel). Taken together, these results suggest that the ASH response may be mediated by rapid calcium influx via an unidentified type of calcium channels and by slow influx via EGL-19 L-type VGCCs, both of which are amplified by CICR. In previous studies, ASH response was significantly suppressed by NemA-treatment as well as by the *egl-19(n582)* mutation upon stimulation with another repulsive odorant (1-octanol) (*Zahratka et al., 2015*) although ASH response was essentially unaffected by the same *egl-19* mutation upon stimulation with glycerol (*Pokala et al., 2014*), suggesting that different sensory stimuli may be processed differently in the polymodal ASH neurons.

## Discussion

In this study, we quantitatively analyzed olfactory behavior in *C. elegans* and found that the animals appear to choose the appropriate migratory direction efficiently on the gradient of a repulsive odor based on subtle changes in odor concentration. This result suggests a novel behavioral strategy for navigating sensory gradients (*Figure 1E*). One of the problems in analyzing animals' responses in a traditional behavioral arena is that the behavior of animals occurs in a closed loop configuration, *i.e.*, the change in movement directly feeds back onto sensory input. Therefore, when the changes in sensory stimulus and behavior are simultaneously observed, it is difficult to distinguish whether sensory input or behavior is the cause of the change. To solve this problem, we established a rigorous quantitative platform to understand the sensory behavior of *C. elegans*: We measured the temporal changes in sensory stimulus in a traditional behavioral paradigm and reproduced it in an integrated microscope system (the OSB2 system) in an open loop configuration. This made it possible to quantitatively measure behavioral responses caused by sensory stimuli, and the neural activities mediating them.

Based on the results obtained in our system, we consider that the olfactory behavior of the animals comprises two essential features of decision-making at the behavioral level. (1) Discrete behavioral choice based on sensory information: Based on *dC/dt*, *C. elegans* choose either a high-turning state (pirouette) or low-turning state (run), which were shown to be physiologically distinct (*Figure 2D* and *Figure 1—figure supplement 1A and E*). (2) Timing of decision: The time required for the transition from a high-turning state to a low-turning state is shorter when the odor concentration decreases rapidly, and is longer when the concentration decreases slowly (*Figure 4B*). Moreover, we found that $[Ca^{2+}]_i$ is accumulated in AWB neurons according to a leaky integrator equation of odor concentration change, which is correlated with the behavioral transition (*Figure 4B* and *Figure 4—figure supplement 2*). Because odor concentration is a critical piece of information for the choice of behavioral states, the temporal integration in AWB neurons can be regarded as the 'evidence accumulation' for decision-making.

### Transition between two behavioral states is regulated by different computations of odor concentration

The behavioral transitions in 2-nonanone avoidance are regulated by ASH neurons for odor increase and AWB neurons for odor decrease (*Figure 8A*). ASH neurons respond to unfavorable changes in the odor concentration in a time-differential manner and initiate turns rapidly. Such a 'reflex'-like response is consistent with the nociceptive features of ASH neurons and their synaptic connectivity, whereby the animals respond to various noxious stimuli by inducing turns and reversals (*Bargmann, 2006*; *Kaplan, 1996*). Time-differential properties have been suggested for the transient responses of ASH and other sensory neurons with all-or-none stimuli in *C. elegans* (*Lockery, 2011*). This property has also been reported in a recent study of *Drosophila* olfactory neurons (*Schulze et al., 2015*) and is in agreement with the general idea that dynamic signal changes are mediated by phasic sensory receptors (*Delcomyn, 1998*). Mathematically, the ASH activity can also be approximated by the same leaky integrator equation as AWB, whose fitness is higher than the time-differential equation according to the Bayesian Information Criterion (BIC) (*Figure 4—figure supplement 3* and *Table 3*). This is consistent with the recent report that ASH neurons temporally integrate sensory signals over several seconds (*Kato et al., 2014*). However, the onset of responses

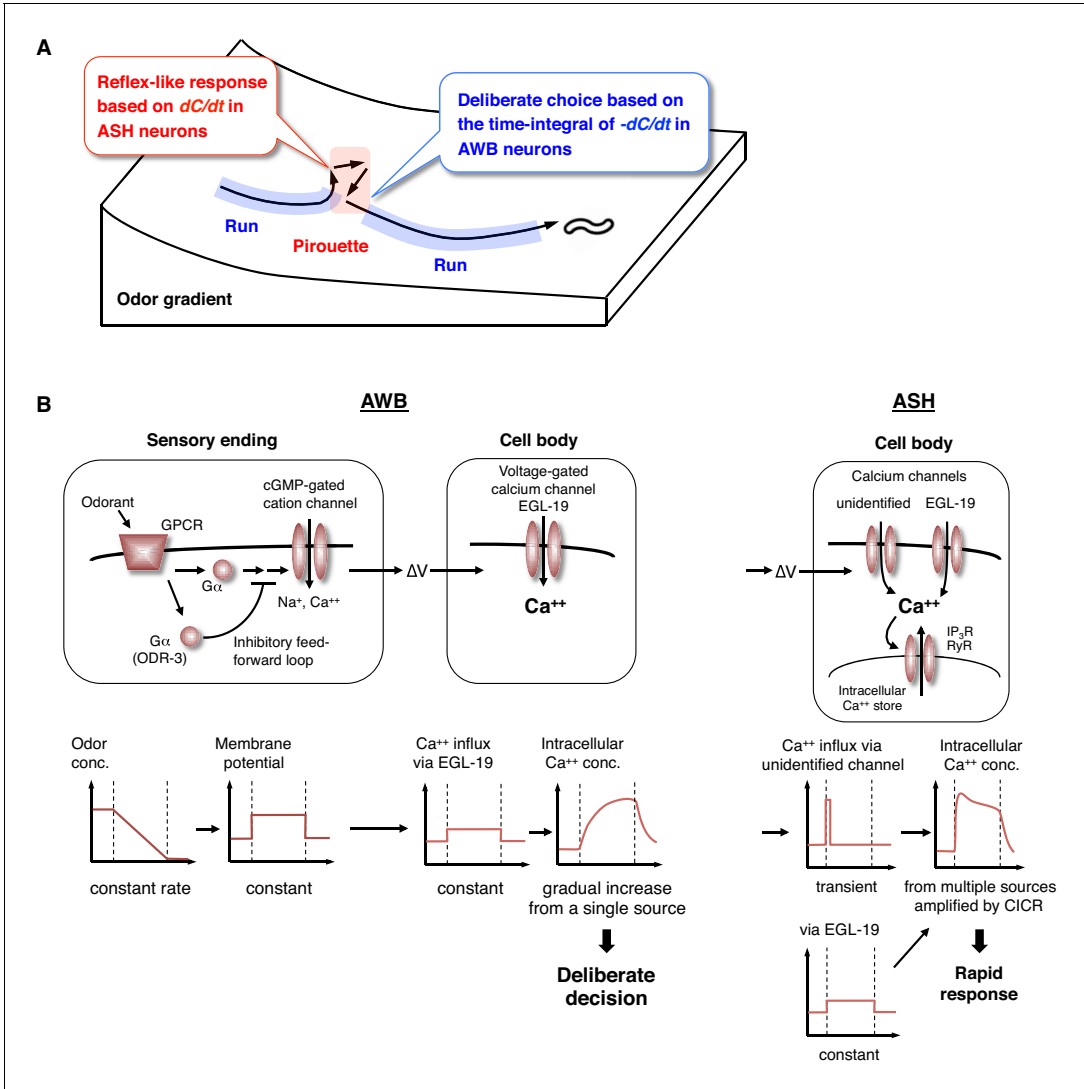

**Figure 8.** Physiological and molecular models of decision-making by *C. elegans* during odor avoidance. (**A**) Computations of ASH and AWB neurons during odor avoidance behavior. (**B**) Model of the molecular mechanisms for temporal computation of odor information in AWB and ASH neurons. (Left) In AWB neurons, odor decreases likely cause the activation of Gα proteins as an odor-OFF response (**Bargmann, 2006**; **Usuyama et al., 2012**), where an unidentified Gα positively transduces the signal and ODR-3 inhibits the signal for the time-differential computation. The Gα signaling is transmitted to the cGMP-gated cation channel TAX-2/TAX-4 (**Bargmann, 2006**) to cause depolarization. Depolarization then triggers calcium influx via EGL-19 at the cell body, which causes the gradual accumulation of $[Ca^{2+}]_i$. (Right) In ASH neurons, the depolarization at the sensory ending triggers an unidentified rapid and transient calcium channels, as well as EGL-19. The calcium influx through these channels is amplified by CICR via RyR (UNC-68) and $IP_3R$ (ITR-1).

was not sufficiently reproduced with the model (*Figure 4—figure supplement 3*, red arrows). Our result in *Figure 7D* suggests that the ASH response consists of the time-integral component mediated by EGL-19 L-type VGCCs and the fast and transient component, possibly mediated by as-yet-unidentified channels. Nociceptive ASH neurons may have developed specialized mechanisms to quickly cause aversive responses based on a slight change in the undesirable signal.

In contrast, AWB neurons extract long-term information about favorable changes in odor concentration using the time-integral property, which leads to 'deliberate' transitions from pirouettes to runs. This time-integral property has not been characterized previously in the sensory response of *C. elegans*. The sensory responses of the animals to gradual signal changes have been reported in a few cases, although they were time-differential, stochastic or tonic, rather than time-integral

(*Biron et al., 2008*; *Kimura et al., 2004*; *Luo et al., 2014*). In reality, sensory signals likely change with noise rather than at a constant rate. Moreover, an animal's movement itself causes fluctuations in the sensory input. Thus, it is reasonable to conclude that animals extract sensory information over a longer time window by using a time-integral property, such as that described here for AWB neurons. To trigger behavioral transitions according to neuronal activity, a threshold for temporal integration is required. AIZ interneurons, the major postsynaptic targets of AWB neurons (*White et al., 1986*), have been shown to elicit digital-like excitability and trigger turns upon activation (*Li et al., 2014*), suggesting that AWB neurons inactivate AIZ neurons in an all-or-none manner at a certain threshold.

## Molecular mechanisms of the temporal integration of sensory information

Through genetic analyses, we investigated the molecular mechanisms underlying the computations in AWB and ASH neurons. First, we found that the temporal integration of sensory information occurs within the AWB neuron itself and is not dependent on synaptic connections. To the best of our knowledge, this result is the first experimental demonstration of evidence accumulation for decision-making within a neuron (see below). We also found that the ODR-3 G$\alpha$ protein possibly plays a time-differential role in the sensory signaling of AWB neurons. The result was surprising because ODR-3 has been considered to play a major stimulatory role in sensory signal transduction, not in its inhibitory modification (*Bargmann, 2006*; *Roayaie et al., 1998*). Because ODR-3 is homologous to the inhibitory G$\alpha$i proteins (*Bargmann, 2006*), it is possible that the sensory signal via another unidentified G$\alpha$ might be downregulated by the activity of ODR-3—such an inhibitory feed-forward loop could compute the time-differential of the input (*Figure 8B*, left) (*Alon, 2007*). In ASH neurons, ODR-3 appears to transduce the sensory signal itself (likely with other G$\alpha$); the different roles of ODR-3 in AWB and ASH neurons may be related to the fact that AWB and ASH neurons use different molecular pathways for signal sensation (*Bargmann, 2006*).

Together with the finding that the time-integral calcium increase in AWB neurons is mainly mediated by EGL-19 L-type VGCCs, these results suggest the following model (*Figure 8B*, left). A constant change in odor concentration leads to a constant and persistent depolarization with a time-differential activity of ODR-3 G$\alpha$ at the sensory ending. This depolarization is conducted to the cell body and causes the constant activation of EGL-19 L-type VGCCs and the constant influx of extracellular calcium, which leads to the gradual accumulation of $[Ca^{2+}]_i$. In agreement with this model, L-type, but not N/P/Q-type or T-type, VGCCs are known to be activated continuously during depolarization (*Hille, 2001*). This gradual $[Ca^{2+}]_i$ accumulation causes a temporal delay and deliberate decision, acting as a low-pass filter to smooth out noisy sensory inputs. By contrast, in ASH neurons, the depolarization causes the activation of EGL-19 L-type VGCCs and other type(s) of rapid and transient calcium channels, where a small influx of calcium leads to its amplification with CICR and to a rapid behavioral response (*Figure 8B*, right).

In principle, temporal integration of sensory information for behavioral choice in *C. elegans* during odor avoidance is analogous to evidence accumulation in perceptual decision-making in mammals. In both cases, animals extract a long-term trend in sensory information by temporally integrating the information for decision-making (*Gold and Shadlen, 2007*; *Schall, 2001*). While AWBs are sensory neurons, their integration property may reflect a central nervous system-like function; *C. elegans'* sensory neurons are known to possess higher-order functions, such as learning and memory, because of the quite small number of neurons in their nervous system (*de Bono and Maricq, 2005*; *Sasakura and Mori, 2013*). In higher animals, temporally integrating neural activity is required for decision-making as well as for other brain functions, such as working memory, and is generally considered to be mediated by recurrent neural circuits (*Gold and Shadlen, 2007*; *Schall, 2001*; *Wang, 2008*). This temporally integrating activity has also been suggested to be mediated by intracellular calcium signaling within a cell (*Curtis and Lee, 2010*; *Loewenstein and Sompolinsky, 2003*; *Major and Tank, 2004*); however, this had not been demonstrated experimentally. We speculate that single-cell temporal integrators with L-type VGCCs, such as the AWB neurons, may also be involved in decision-making and working memory in higher animals. Interestingly, the odor avoidance behavior of *C. elegans* is regulated by dopamine (*Kimura et al., 2010*), a neuromodulator involved in mammalian decision-making (*Schultz, 2007*). Thus, further genetic analysis of the

behavioral choice during odor avoidance behavior in *C. elegans* may identify previously undescribed evolutionarily conserved molecular mechanisms that are responsible for decision-making.

## Materials and methods

### Strains

The techniques used for culturing and handling *C. elegans* were essentially as described previously (*Brenner, 1974*). The *C. elegans* wild-type Bristol strain N2, RRID:WB-STRAIN:JD21 *cca-1(ad1650)*, RRID:WB-STRAIN:MT1212 *egl-19(n582)*, RRID:WB-STRAIN:JT73 *itr-1(sa73)*, RRID:WB-STRAIN: CX3222 *odr-3(n1605)*, RRID:WB-STRAIN:CX2205 *odr-3(n2150)*, RRID:WB-STRAIN:CB55 *unc-2(e55)*, RRID:WB-STRAIN:MT7929 *unc-13(e51)*, RRID:WB-STRAIN:DA509 *unc-31(e928)*, and RRID:WB-STRAIN:CB540 *unc-68(e540)* were obtained from the Caenorhabditis Genetics Center (University of Minnesota, USA). In all the behavioral and physiological experiments, young adult hermaphrodites were used.

### Multi-worm tracking and analysis of 2-nonanone avoidance on a 9 cm plate

Quantitative analysis of the 2-nonanone avoidance of wild-type animals in the 9 cm agar plate was carried out as previously described (*Kimura et al., 2010*; *Yamazoe-Umemoto et al., 2015*). In brief, several adult animals per assay were transferred to the center of a 9 cm NGM agar plate either directly from a standard 6 cm nematode growth medium (NGM) plate with the food bacteria OP-50 ('fed') or after a 1 hr starvation on the NGM plate without OP-50 ('starved'). In the following analysis, we used a data set that is a mixture of 50 fed and 50 starved animals. Although we did not find a significant difference between fed and starved animals in 2-nonanone avoidance (*Kimura et al., 2010*), the feeding state (fed or starved) could affect some aspects of *C. elegans*' behavior (*Bargmann, 2006*) and we wanted to focus on feeding state-independent behavioral aspects of the animals. Two μL of 30% 2-nonanone (diluted in EtOH) were put in two spots on the surface of the agar plate (*Figures 1A* and *2A*), and images of the animals during the avoidance behavior were captured at 1 Hz for 12 min by our multi-worm tracking system with a high-resolution camera in a fixed position (*Kawazoe et al., 2013*; *Yamazoe-Umemoto et al., 2015*). In this study, we used a CMOS camera CSB4000F-10 (Toshiba Teli Corp., Japan) equipped with a C mount adaptor and a Nikkor 50 mm f/1.2 lens (Nikon Corp., Japan). Because the camera captures the entire area of the 9 cm plate with a resolution of 2008 × 2044 pixels, an animal of length ~1 mm and width ~0.05 mm is depicted in ~25 pixels. $x−y$ coordinates of the centroids of the animals in each image were measured by Move-tr/2D software (Library Inc., Japan), and were further analysed by Excel2010 (Microsoft) or R (The R Project). Because the animals did not initiate avoidance during the first 2 min on average (*Kimura et al., 2010*) (*Figures 1A* and *2B* and *Figure 2—figure supplement 1E*), data between 121–720 s were used for the analysis (*Tanimoto et al., 2017*).

### Definition of pirouettes and runs

A pirouette is a period of frequent turns and migrations whose duration is shorter than a threshold value (*Pierce-Shimomura et al., 1999*). The animal's behavioral state in one second was classified as a turn if the absolute value of angle change in migratory vector of the animal's centroid from the previous second (*i.e.*, during 1 s) was larger than 90° or if the migratory velocity was smaller than 0.1 mm/s in the following frames after the large angle change. According to this definition, the reverse and the omega turn (*Gray et al., 2005*) were recognized as turns. A distribution of turn intervals (*i.e.*, migratory durations) during 2-nonanone avoidance was well-fitted by a sum of two exponentials for shorter and longer intervals (*Figure 1—figure supplement 1A*). A period at which the numbers of the short and long intervals were equal was 13.1 s and determined as $t_{crit}$ according to the original definition (*Pierce-Shimomura et al., 1999*). Migrations whose turn interval was longer than $t_{crit}$ were classified as runs, and migrations shorter than $t_{crit}$ as well as turns were classified in pirouettes.

### Bearings at the initiation of and during runs

The directions of animal migrations for 1 s were defined in terms of the bearing, *B*, with respect to the 2-nonanone gradient, where *B* = 0° indicates migration directly away from the odor source (*i.e.*,

down the gradient) and $B = \pm180°$ indicates migration directly toward the odor source (*i.e.*, up the gradient). Bearing at run initiation in salt-taxis by the previous study was calculated from the results of taxis toward $NH_4Cl$ (56.0%) and biotin (55.2%) in Figure 9 of the report (*Pierce-Shimomura et al., 1999*).

### Calibration curve for 2-nonanone measurement

Calibration curve for 2-nonanone measurement is described in more detail at Bio-protocol (*Yamazoe-Umemoto et al., 2018*). To measure local concentrations of gaseous 2-nonanone in the assay plate, we used a gas chromatograph (GC) with a sensitive semiconductor detector, SGVA-N2, which was optimized for 2-nonanone detection (FIS Inc., Japan). To make a calibration curve for the measurement, 0.36, 1.07, 3.56, 35.6, 59.4, 97.2, and 200 μL of liquid 2-nonanone (Wako Pure Chemical, Japan) were vaporized in a 50 L tank DT-T1 (FIS Inc.), each corresponding to 0.04, 0.12, 0.4, 4.0, 6.8, 11.1, and 22.9 μM in the gas phase, respectively. After the volatilization period, 0.2 mL of the gas was sampled with a 2 mL plastic syringe with a needle from an outlet of the tank and was immediately injected into the GC. The volatilization periods were determined for each amount of the liquid to maximize the 2-nonanone signal. Synthetic air Alphagaz 1 (Air Liquide, Japan) was used as a carrier gas. With 260 s retention time, a single large peak of signal intensity (mV) was detected as 2-nonanone signal (*Figure 2—figure supplement 1B*). The experiments were repeated 3–4 times for each concentration. The correlations between the peak height of the signal and the gaseous 2-nonanone concentration in a log-log plot were well-fitted by two simple regression lines for lower and higher concentrations (*Figure 2—figure supplement 1C*; $R^2$ >0.999 for both). In general, for semiconductor detectors, the correlation between the peak height of the signal and signal concentration in a log-log plot are well-fitted by two simple regression lines for lower and higher concentrations.

### Measuring odor gradient by gas chromatograph

Measuring odor gradient by gas chromatograph is described in more detail at Bio-protocol (*Yamazoe-Umemoto et al., 2018*). For the odor sampling, a hole of 1 mm in diameter was made through the bottom of the plastic plate and the agar. Because the molecular weight of 2-nonanone (FW 142.2) is larger as a volatile compound, it did not leak easily from such a small hole. 1, 3, 6, 9 and 12 min after placing the odor at the two spots, a 2 mL plastic syringe, which is the same type as the one used in the calibration, was inserted into the plate through the hole from the bottom, and 0.2 mL of the gas phase was sampled (*Figure 2—figure supplement 1A*). Each plate was used only once to avoid disturbance of the gradient by the sampling. The sampled gas was immediately injected into the GC for measurement. The concentration of 2-nonanone was calculated from the height of the signal peak according to the regression line for the calibration. For each data point, the measurements were repeated 7–9 times and median and quartile was calculated for the fitting.

### Fitting the odor gradient and calculation of $C_{worm}$

Fitting the odor gradient and calculation of $C_{worm}$ are described in more detail at Bio-protocol (*Yamazoe-Umemoto et al., 2018*). The least squares method was used to fit the measured concentration. In the closed plate, the odor concentration asymptotically approaches a constant value. Therefore the measured concentrations were fitted to a phenomenological curve with two exponential saturation functions: $C(x, y, t) = a(r_1)(1-\exp(-b(r_1)t)) + a(r_2)(1-\exp(-b(r_2)t))$. $r_1$ and $r_2$ are the distances from the position $(x, y)$ on the agar to the two odor sources. The asymptotic concentration $a(r)$ and the increasing rate $b(r)$ are functions of the distance $r$ such as $a(r)=a_0 \exp(-a_1r - a_2r^2)$ and $b(r)=b_0 \exp(-b_1r - b_2r^2)$. The assumption that $C(x, y, t)$ is given by the sum of the two independent functions is valid for the low concentration regions $x > 0$. The fitting parameters $a_0 = 20.68$ μM, $a_1 = 0.7355$ cm$^{-1}$, $a_2 = -0.05408$ cm$^{-2}$, $b_0 = 0.8384$ min$^{-1}$, $b_1 = 0.7835$ cm$^{-1}$ and $b_2 = -0.05761$ cm$^{-2}$ were determined by the Levenberg-Marquardt method (*Press et al., 1992*). We consider the measured and the fitted odor gradient as reliable because it is consistent with the fact that the amount of 2-nonanone at the source was apparently reduced by 20–30% after 12 min and with a theoretically calculated simulation (*Yamazoe-Umemoto et al., 2015*). The 2-nonanone concentration at a given temporal and spatial point of an animal's centroid was calculated from the fitting curve and was designated as $C_{worm}$. Turning rate shown in *Figure 2D* was determined as the relationship

between the $dC_{worm}/dt$ during one second of migration and the probability of turning in the next second.

## Molecular biology and germline transformation

For the cell-specific expression of mCherry (*Shaner et al., 2004*), GCaMP3 (*Tian et al., 2009*), ChR2 (C128S) (*Berndt et al., 2009*) and Arch (*Chow et al., 2010*), *str-1* (*Troemel et al., 1997*) or *srd-23* promoter (*Colosimo et al., 2004*) was used for AWB-expression, and *sra-6* promoter (*Troemel et al., 1995*) was used for ASH-expression. Germline transformation was performed using microinjection (*Mello et al., 1991*). The plasmids and strains used in this study are listed in *Supplementary file 2* and *3*. In *Figures 3*, *4* and *6B* and *Figure 6—figure supplement 1*, multiple transgenic lines were used for each type of experiment, and the different lines produced similar results. The representative transgenes (*i.e.*, extra chromosomal arrays) were used for genetic analyses in *Figures 6A* and *7*.

## Behavioral tracking with the integrated microscope system

For the OOSaCaBeN (*O*lfactory and *O*ptogenetic *St*imulation *a*ssociated with *Ca*lcium imaging on *Be*having *N*ematode, or OSB2) system, we integrated an auto-tracking microscope system for calcium imaging and optogenetic manipulation with an odor-delivery subsystem (*Busch et al., 2012*; *Tanimoto et al., 2016*). Briefly, a wild-type *C. elegans* (N2) on a NGM plate was placed on a motorized stage HV-STU02 (HawkVision, Japan) combined with an upright microscope and illuminated with infrared light. Bright field images of the animal were acquired by a charge-coupled device (CCD) camera at 200 Hz to regulate the motorized stage for maintaining the region-of-interest (ROI) of a freely moving animal in the center of the view field of the microscope ('ROI-tracking', *Video 2*) (*Maru et al., 2010*). A ROI was set around the head neuropil. The system also allowed us to maintain the centroid of a whole animal in the center of the view field ('centroid-tracking', *Video 4*). ROI-tracking was used for calcium imaging with a 20× objective lens, and centroid-tracking was used for wild-type behavioral analyses and optogenetic behavioral analyses with a 10× objective lens. The behaving animal was continuously exposed to an odor flowing from two syringe pumps (*Video 3*), which changed the odor concentration according to a predefined program.

## Odor delivery

For odor delivery, 4 μM of 2-nonanone was sampled from the 50 L vaporizing tank with a 25 ml Gas-tight Syringe (Hamilton, USA). Two such syringes were set on a syringe pump HV-SSP01 (HawkVision, Japan) that was controlled by the same program for the auto-tracking. Adapting the gas delivery strategy described previously (*Busch et al., 2012*), one pump was used for 2-nonanone and the other one was for air. The pump speeds were programmed to deliver a constant gas flow of 8 mL/min from the end of the tube, but with varying combinations from each pump to make the temporal gradient of 2-nonanone concentration. For example, when the pump speed of 2-nonanone syringes was changed from 2 ml/min to 5 ml/min, the air syringes went from 6 ml/min to 3 ml/min during the same period. The programs of the pump speeds were designed so that the magnitude of $dC/dt$ was similar to that which animals experienced during the odor avoidance assay in the plate (*Figure 2*). The actual concentration of 2-nonanone was monitored at the end of the tube by the same type of semiconductor sensor as the one in the gas chromatograph (GC), and the values were recorded with a PC via a digital multimeter MAS345 (Mastech, Hong Kong) before and after the behavioral assays for each day. The sensor was calibrated every day with a similar method as the GC, with calibration concentrations of 0.5, 1, 2, and 4 μM.

We consider that the measured odor concentrations (*Figures 3*, *4*, *6* and *7*) closely matched to the actual odor concentration that the animals experienced during the odor avoidance behavior (*Figures 1* and *2*) because of the following reasons. (1) The tube end was always maintained at ~1 mm from the freely-moving animal during tracking, and the entire body of an animal was exposed to essentially uniform odor flow without significant diffusion and/or turbulence (*Figure 3—figure supplement 1A* and *Video 3*). (2) With the flow (8 ml/min), the animals exhibit robust behavioral response reproducibly through multiple trials (*Figures 3* and *4*) while the flow itself did not affect the animal's behavior. (3) The animals likely sense the odor concentration in air phase but not in

water phase (*i.e.,* agar surface) because of the high hydrophobicity of 2-nonanone (a nine carbon ketone).

## Quantitative behavioral experiments on the OSB2 system

The behavior of animals was calculated from records of displacement of the auto-tracking stage and from the position of the ROI or centroid of the view field. For ROI-tracking, the trajectory of an animal's behavior was wavier than for centroid tracking because the ROI was usually set around the animal's head, which moves in a sinusoidal pattern (*Video 2*). To compensate for the wavy pattern, the $x - y$ coordinates for ROI-tracking were calculated as a moving-average for ±10 frames at 10 Hz (*i.e.,* ±1 s). This gave similar results to centroid-tracking on the quantitative behavioral analysis. The migratory trajectory from either of the tracking methods was sampled at each second (1 Hz), and a change in the migratory vector for 1 s larger than 90° was recognized as a turn. In the Figures, ensemble averages in each 10 s bin are shown.

In *Figure 4A and B*, we investigated the time when the turning rate changed based on the rate of increase or decrease in odor concentration. In order to investigate the timing, it was necessary to finely set the time window. However, since a turn is an uncommon occurrence (a turning rate of 0.1 is once in 10 s), narrowing the time window increased the variation. In order to obtain the same number of turns as the 60 s using a time window of 10 s, six times as much sampling had to be performed. Furthermore, even more samples were required for performing multiple tests. Therefore, we used the prediction interval, a criterion in the field of statistical inference. The 99% prediction interval is an interval in which future data will fall with 99% probability, if it obeys the same probability distribution as the previously observed data (in this case, odor-zero or odor-plateau phase). A $100(1-\alpha)$% prediction interval on a single future observation $(X_{n+1})$ from a normal distribution is given by the following formula:

$$\bar{x} - t_{\frac{\alpha}{2},n-1} s \sqrt{1 + \frac{1}{n}} \leq X_{n+1} \leq \bar{x} + t_{\frac{\alpha}{2},n-1} s \sqrt{1 + \frac{1}{n}}$$

where $\bar{x}$ is the sample mean, $n$ is the number of previously observed data, $t_{\alpha/2,n-1}$ is the $100(1-\alpha/2)$ percentage point of a *t*-distribution with $n-1$ degrees of freedom, and $s$ is the sample standard deviation (*Montgomery and Runger, 2002*). Using this criterion, we analyzed 'timing when the unexpected value appears for the first time' in *Figure 4A and B*.

## Calcium imaging

The details of calcium imaging with the OSB system were previously described (*Tanimoto et al., 2016*). In brief, the sample was exposed to excitation light from a MiLSS (Multi-independent Light Stimulation System, Aska Company, Japan) (*Sakai et al., 2013*). The images for GCaMP and mCherry were split and simultaneously captured side-by-side on an EM-CCD camera ImagEM with W-View system (Hamamatsu, Japan). Images were taken at a 32.6 ms exposure time and 100 ms sampling interval with 2 × 2 binning. The cell body was tracked off-line with another custom-made program for the centering (*Video 2*), and signal intensities of particular regions were measured by ImageJ (NIH). The data from frames where the cell body was not centered were omitted. The signal intensity of the background was subtracted from that of the cell body, and the value was moving-averaged for ±1 frames and further analysed. The average of fluorescence intensity of GCaMP during 1 min before the odor increment or decrement was defined as the baseline $F_0$. Because $\Delta F/F_0$ of GCaMP and the ratio between fluorescence intensities of GCaMP and mCherry (GCaMP/mCherry: $R$) exhibited similar tendencies, and because and the noise level was smaller in $\Delta F/F_0$ than in $R$, the data of $\Delta F/F_0$ were used in the figures. In *Figure 7*, $\Delta R$ was used because the mutations in *itr-1* or *unc-68* could affect the baseline as well as the response calcium levels of the neurons. Also in *Figure 7*, the animals were immobilized with the acetylcholine receptor agonist levamisole for high-throughput analysis, in which multiple animals were stimulated and imaged simultaneously. Even with the levamisole treatment, the responses of AWB and ASH neurons in the naive wild-type animals were essentially similar to those in the freely moving animals (*Figure 4A and B*).

## Optogenetic analysis

Animals were raised in the presence or absence of ATR according to the previous report (*Kawazoe et al., 2013*), and transferred to an NGM plate on the OSB2 system and maintained under the objective lens by auto-tracking. For ChR2(C128S) experiments in the absence of a 2-nona-none stimulus (*Figure 3D*), after 1 min without light stimulation, the animal was transiently illuminated with blue light (3 s) for activation through BP460-495 and DM505 with ND25 (~0.8 mW/mm²). Turning rates of 30–60 s and 65–95 s were calculated as before or after the blue light illumination, respectively. The turns of 60–65 s were not included in the calculation because blue light illumination (60–63 s) appeared to somewhat affect the animals' locomotion for a few seconds (*Ward et al., 2008*). For Arch experiments in the presence of a 2-nonanone stimulus (*Figures 3F* and *4C*), green light was delivered through BP530-550 and DM570 at ~1.0 mW/mm², and turning rates were calculated. The optical filters were from Olympus.

## Mathematical modeling of neuronal responses

For the time-differential models of neuronal responses, the following time-differential equation was used:

$$X(t) = k\frac{dC(t)}{dt}$$

where $X(t)$ is neuronal response, $k$ is the conversion factor, and $C(t)$ is the measured odor concentration. The $dC(t)/dt$ was calculated as the central difference of $C(t)$. This equation indicates that the neuronal response $X(t)$ responds to the odor gradient $dC(t)/dt$ at each time. The value of $k$ was determined by the least squares method to fit $X(t)$ to the measured $\Delta F/F_0$ in response to the odor gradients.

For the time-integral models of neuronal responses, the following leaky integrator equation was used:

$$\frac{dX(t)}{dt} = kI(t) - \frac{1}{\tau}X(t)$$

where external input was given by temporal odor change; $I(t) = dC(t)/dt$. $\tau$ is the time constant of leaky integration. $k$ and $\tau$ were determined by the least squares method to fit $X(t)$ to the measured $\Delta F/F_0$ responded to the odor gradients. This differential equation was numerically integrated by the Euler method with a time-step of 1 s. The initial value was $X(t) = 0$ which corresponds to $\Delta F/F_0 = 0$ in the basal state. For *odr-3* mutants, on the other hand, external input was $I(t) = -(C(t) - C(t - \Delta t))/\Delta t$ in the leaky integrator equation, and the values $k, \tau$, and $\Delta t$ were determined to fit $X(t)$ to the measured $\Delta F/F_0$ of *odr-3*.

Estimation of intracellular calcium concentration in *Figure 4—figure supplement 2* was conducted as follows: Since the relationship between fluorescence signals and calcium concentration is non-linear, a change in the neuronal activity to stimulation is properly evaluated, not by the fluorescence intensity of the calcium indicator, but by the calcium concentration itself. Taking the non-linear relationship into account, intracellular calcium concentration $[Ca^{2+}]$ was estimated by the Hill equation; $(F - F_{min})/(F_{max} - F_{min}) = [Ca^{2+}]^h/([Ca^{2+}]^h + K_d{}^h)$. The $F$ is the measured fluorescence intensity, $F_{min}$ and $F_{max}$ are the fluorescence intensities under $Ca^{2+}$-free and $Ca^{2+}$-saturated conditions, respectively. The $h$ is the Hill coefficient and $K_d$ is the dissociation constant. For GCaMP3, the values of $h$ and $K_d$ were reported previously (*Akerboom et al., 2012*). In each experiment, $Ca^{2+}$ response to stimulation is expressed as the ratio of the fluorescence response to the basal fluorescence intensity $F_0, \Delta F/F_0 = (F - F_0)/F_0$. By solving the Hill equation for $[Ca^{2+}]$ in terms of the fluorescence intensities, the following equation to calculate the intracellular calcium concentration from the measured ratio $\Delta F/F_0$ was obtained:

$$[Ca^{2+}] = K_d\left(\frac{1 + \Delta F/F_0 - f_{min}}{f_{max} - 1 - \Delta F/F_0}\right)^{1/h}$$

where $f_{min} = F_{min}/F_0$ and $f_{max} = F_{max}/F_0$ are the minimum and maximum fluorescence intensities relative to $F_0$, respectively. For GCaMP3, $f_{max} = 12f_{min}$ since the dynamic range $F_{max}/F_{min}$ (i.e., $f_{max}/f_{min}$) is

reported to be ~12 fold (*Tian et al., 2009*). The time delay of fluorescence response to a calcium concentration change was not taken into account since the temporal resolution of the odor concentration measurement was of the second order, while the association and dissociation time constants of GCaMP3 are of the sub-second order (*Tian et al., 2009*). When $X(t)$ corresponds to the calcium concentration, the basal value of $X(t)$ in the steady state is not zero since the intracellular calcium concentration is not reduced to zero even in the basal state. Therefore, the leaky integrator equation was generalized as follows:

$$\frac{dX(t)}{dt} = kI(t) - \frac{1}{\tau}(X(t) - X_{base})$$

where $X_{base}$ corresponds to the basal calcium concentration in the steady state and takes a positive value. For AWB and ASH neurons, unknown model parameters $k, \tau, f_{min}$ and $X_{base}$ were determined by the least squares method to fit $X(t)$ calculated by the generalized leaky integrator equation to the calcium concentrations estimated from $\Delta F/F_0$. Similar estimation of non-linear property of GCaMP3 has been reported previously (*Kato et al., 2014*). The time-differential and time-integral models reasonably approximated the neural responses under the conditions used in this study. However, with stronger odor concentration changes, input saturation may need to be considered, in which case the input could be put through a logistic sigmoid function for example. The values of the fitting parameters are shown in *Tables 1*, *2*, *3* and *4*.

## Computer simulation of 2-nonanone avoidance behavior

The previous algorithms (*Iino and Yoshida, 2009*; *Yamazoe-Umemoto et al., 2015*) were modified as follows to simulate 2-nonanone avoidance behavior (*Figure 5*). The parameters for simulation were based on the migratory statistics of real wild-type animals and contained no free parameters unless otherwise indicated. The model animal moved at a speed of 0.14 mm/s. In the low-turning state, the model animal moved forward with fluctuations in migratory direction, which was randomly chosen from the Gaussian distribution of −0.065 ± 5.14° (mean ± SD). The odor signal periodically fluctuated because of the sinusoidal movement of the animal. The position of animal's anterior end, where the sensory endings of ASH and AWB neurons are located, was calculated as a sine curve along the animal's track. The amplitude and frequency of the sine curve was 0.1 mm and 0.5 Hz, respectively (*Kimura et al., 2004*; *Shen et al., 2012*). The track of the anterior end was used for the calculation of $C_{worm}$. A turn occurred based on the pirouette initiation rate of 0.0326/ (0.200 + exp (−231 × $dC/dt$))+0.0260, which is relatively constant (~0.03 $s^{-1}$) when $dC/dt$ <0 and increases when $dC/dt$ >0; The $dC/dt$-dependency in the pirouette initiation rate was determined from the probability of pirouette initiation after 2 s of the step for real animals. The turning duration was 3 s. After a turn, the model animal was in the high-turning state and initiated a migration, whose deviation in direction from the direction just before the turn was randomly chosen from a pool of the measured values in real animals. In the high-turning state, the model animals turned at a constant rate of 0.2 $s^{-1}$, which results in ~95% of migratory duration shorter than the threshold value $t_{crit}$ (13.1 s). Therefore, in the high-turning state, most of the migrations were classified as pirouettes. When the model animals happened to migrate down the gradient and experienced $dC/dt$ <0, their state transited from high- to low-turning according to the leaky integration of $dC/dt$ described in the previous section. When the leaky integration of $dC/dt$ became higher than 0.18, the high turning state was switched to the low turning state. The threshold value 0.18 was chosen as a value similar to the one associated with the turn suppression in *Figure 4B*. For the 'differentiation model,' $dC/dt$ itself was used for the initiation of a low turning state instead of leaky integration, and the high turning state was switched to the low turning state when $dC/dt$ was negative. The simulation was repeated 100 times for each model animal condition. Time was discretized into intervals with $\Delta t$ = 1 s.

## Data analysis and statistics

For the experiments with the OSB2 system, the data were obtained on multiple days from approximately 20–50 animals for each condition. We chose this sample number because a large scale behavioral analysis of *C. elegans* concluded that 20 animals would discriminate single SD in a behavioral phenotype at over 80% power, and 24 ± 14 (average ± SD) animals per condition were used in the study (*Yemini et al., 2013*). For *Figures 1* and *2* and *Figure 1—figure supplements 1, 100* animals

were used because we investigated various aspects of behavior in detail. After the sample acquisition, the data of some animals for *Figures 3*, *4*, *6* and *7* were excluded when any of the following problems were found: (1) trials interrupted by errors in auto-tracking, (2) animals with too weak intensity of basal mCherry or GCaMP3 fluorescence for off-line tracking, (3) animals with abnormal sudden transient activation of AWB during odor-zero or odor-up phases (8 out of 279 animals tested for AWB), or (4) animals with basal locomotion speeds slower than 0.02 mm/s (average speed ± SD of normal animals was 0.15 ± 0.04 mm/s). Experimental conditions, such as the presence/absence of ATR, light stimulation, odor gradient, or different strains were randomized on a daily basis.

A Kruskal-Wallis test with a *post hoc* Steel-Dwass test was used for multiple comparisons in *Figures 3B*, *4A, B* and *7C*, while a Mann-Whitney test was used for single comparison in *Figures 1D*, *3D, F* and *4C*, and *Figure 2—figure supplement 1D* (right panel) using R (The R Project) or Prism ver. 5.0 for Mac OSX (GraphPad Software, San Diego, CA). The Mardia-Watson-Wheeler test was used for *Figure 5C* and Watson's $U^2$ test was used for *Figure 3—figure supplement 1B* by using Oriana ver. 3 (Kovach Computing Service, Wales, UK). All the statistical details are shown in *Supplementary file 1*.

The Bayesian information criterion (BIC) was used to assess mathematical model fitting in *Figures 4* and *6*. In BIC, the goodness of fit for the model including a penalty term to discourage overfitting is given by the following equation:

$$BIC = Nln\left(\frac{RSS}{N}\right) + Mln(N)$$

where $N$ is the number of samples (frames) used for the fitting, $RSS$ is the residual sum of squares obtained from fitting by the least squares method, and $M$ is the number of free parameters in a given model, respectively. $M = 1$ for the differentiation model ($k$), $M = 2$ for the leaky integration model ($k$, $\tau$), and $M = 3$ for the leaky integration model for *odr-3* ($k$, $\tau$, $\Delta t$). The lower BIC value means a better fitting with a model.

## Acknowledgements

We thank K Tanaka (FIS Inc., Japan), Y Mori (HawkVision Inc., Japan) and K Ishida (Aska Company, Japan) for generous technical support for the hardwares. We also thank Drs. Y Fujie, I Takeuchi, M Usuyama, M Hendricks, Y Zhang, A Gottschalk, P Sengupta, H Kagoshima, M Koga, S Oda, T Obara, M Chen for materials and technical support, C Bargmann, S Nakanishi, Y Komura, T Kikuchi, M S Kitazawa, O Hobert, K Fujimoto and the Kimura laboratory members for suggestions and comments. Nematode strains were provided by the Caenorhabditis Genetics Center (funded by the NIH Office of Research Infrastructure Programs P40 OD010440), and neuronal information was provided by WormBase (funded by National Human Genome Research Institute grant #41 HG002223) and by WormAtlas (http://www.wormatlas.org).

## Additional information

### Funding

| Funder | Grant reference number | Author |
|---|---|---|
| Japan Society for the Promotion of Science | Grant-in-Aid for JSPS fellows | Yuki Tanimoto Akiko Yamazoe-Umemoto |
| Japan Society for the Promotion of Science | Interdisciplinary graduate school program for systematic understanding of health and disease | Yuki Tanimoto Shuhei J Yamazaki |
| Japan Agency for Medical Research and Development | Brain Mapping by Integrated Neurotechnologies for Disease Studies (Brain/MINDS) | Keiko Gengyo-Ando Junichi Nakai |
| Ministry of Education, Culture, Sports, Science and Technol- | Regional Innovation Cluster Program (City Area Type, | Keiko Gengyo-Ando Junichi Nakai |

| | | |
|---|---|---|
| ogy | Central Saitama Area) | |
| Ministry of Education, Culture, Sports, Science and Technology | KAKENHI JP23115704 | Keiko Gengyo-Ando |
| Ministry of Education, Culture, Sports, Science and Technology | KAKENHI JP21115504 | Junichi Nakai |
| Ministry of Education, Culture, Sports, Science and Technology | KAKENHI JP16H06536 | Junichi Nakai<br>Koichi Hashimoto |
| Ministry of Education, Culture, Sports, Science and Technology | KAKENHI JP23115703 | Koichi Hashimoto |
| Ministry of Education, Culture, Sports, Science and Technology | KAKENHI JP21115502 | Koichi Hashimoto |
| Ministry of Education, Culture, Sports, Science and Technology | KAKENHI JP23115711 | Koutarou D Kimura |
| Ministry of Education, Culture, Sports, Science and Technology | KAKENHI JP16H06545 | Koutarou D Kimura |
| Ministry of Education, Culture, Sports, Science, and Technology | The Osaka University Life Science Young Independent Researcher Support Program | Koutarou D Kimura |
| Japan Society for the Promotion of Science | PRESTO | Koutarou D. Kimura |
| Mitsubishi Foundation | | Koutarou D. Kimura |
| Shimadzu Science Foundation | | Koutarou D. Kimura |
| Takeda Science Foundation | | Koutarou D. Kimura |

The funders had no role in study design, data collection and interpretation, or the decision to submit the work for publication.

## Author contributions

YT, Conceptualization, Investigation, Methodology, Writing—original draft, Writing—review and editing; AY-U, Conceptualization, Investigation, Methodology, Writing—original draft; KF, YK, YM, Investigation, Writing—review and editing; SJY, Validation, Investigation, Writing—review and editing; XF, Software, Investigation, Methodology, Writing—review and editing; KEB, Investigation, Methodology, Writing—review and editing; KG-A, JN, Funding acquisition, Investigation, Methodology, Writing—review and editing; YIi, KH, Software, Funding acquisition, Investigation, Methodology, Writing—review and editing; YIw, Software, Validation, Investigation, Visualization, Methodology, Writing—review and editing; KDK, Conceptualization, Supervision, Funding acquisition, Investigation, Methodology, Writing—original draft, Writing—review and editing

## Author ORCIDs

Yuki Tanimoto, http://orcid.org/0000-0002-7434-678X
Koutarou D Kimura, http://orcid.org/0000-0002-3359-1578

# Additional files

## Supplementary files

• Supplementary file 1. Detailed results of statistical tests in this study are shown.

• Supplementary file 2. Plasmids used in this study are shown.

• Supplementary file 3. Strains used in this study are shown.

**Major datasets**

The following dataset was generated:

| Author(s) | Year | Dataset title | Dataset URL | Database, license, and accessibility information |
|---|---|---|---|---|
| Yuki Tanimoto, Akiko Yamazoe-Umemoto, Kosuke Fujita, Yuya Kawazoe, Yosuke Miyanishi, Shuhei J Yamazaki, Xianfeng Fei, Karl Emanuel Busch, Keiko Gengyo-Ando, Junichi Nakai, Yuichi Iino, Yuishi Iwasaki, Koichi Hashimoto, Koutarou D Kimura | 2017 | Data from: Calcium dynamics regulating the timing of decision-making in C. elegans | http://dx.doi.org/10.5061/dryad.h3680 | Available at Dryad Digital Repository under a CC0 Public Domain Dedication |

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
