## [Decision Letter]

Thank you for submitting your article "Calcium dynamics regulating the timing of decision-making in *C. elegans*" for consideration by *eLife*. Your article has been reviewed by three peer reviewers, and the evaluation has been overseen by a Reviewing Editor and Eve Marder as the Senior Editor. The reviewers have opted to remain anonymous.

The reviewers have discussed the reviews with one another and the Reviewing Editor has drafted this decision to help you prepare a revised submission.

As you see from the attached reviews, all reviewers find your study of interest, but raised a number of critiques ranging from genetics to modeling. We hope that you will be able to address these critiques in a revised manuscript in the next few months. You should scale back your molecular-level conclusions where evidence is not definitive. We look forward to receiving your revised manuscript.

Reviewer #2:

In this study Tanimoto et al. address how sensory information is encoded by neurons to implement behavioral decisions. The authors developed new experimental setups and using these approaches provide an interesting conceptual framework linking sensory neural activity to behavior. More specifically, based on their behavioral data they propose a novel negative chemotaxis strategy during which animals accumulate sensory evidence prior to a behavioral decision, switching from a pirouette to a forward-run behavioral state. By calcium imaging they show that AWB chemosensory neuron responses, in contrast to ASH nociceptive neurons, can be fitted to a leaky-integrator model, which could be an essential feature of the evidence-accumulation mechanism. They substantiate this finding with a computer simulation of chemotaxis.

Investigating the molecular basis for the temporal integration of olfactory stimuli in AWB, they first confirm that the integration is a cell autonomous property of AWB. They then show that the *odr-3* Ga protein is essential for the response property of the neurons.

Furthermore, they show that the calcium accumulation in AWB and ASH can be attributed to L-type Ca VG calcium channels, and not to B-type, thought there must be other components in ASH, as only the late response phase is affected.

This manuscript could present both a technical and conceptual advance showing for the first time that evidence accumulation for decision making can be performed by a single sensory neuron class, including an underlying molecular mechanism, as opposed to circuit mechanisms proposed by work in higher animals. However, I think not all conclusions are sufficiently supported in the present form and some experiments need more repetitions and appropriate statistical tests.

The manuscript is well written and easy to follow with some exceptions that should be addressed in a revised manuscript (see below); the graphs are clear, supplemental data are appropriately provided.

Major comments

1) Experiments in Figure 1–Figure 2 are performed in a closed loop configuration, i.e. movement directly feeds back onto sensory input. This is not the case for the open loop virtual gradient setup in subsequent experiments. This difference should be better addressed in the discussion text.

2) Figure 4 and E. I am not convinced that AWB and ASH sensory response profiles could not be fitted equally well to the alternative model in each panel. The authors should provide goodness of fit results for all models on each neuron dataset and perform appropriate statistical tests showing that one model performs significantly better than the corresponding null hypothesis model.

3) The authors use a threshold to determine when sensory information leads to an abrupt change in turning frequency. It is not sufficiently explained how exactly this 99%-prediction-interval is calculated. Provide more information in main text about the logic and details in Materials and methods section.

4) I am not convinced that the turning rate profiles in Figure 4 middle and right panels show discrete transitions. In Figure 4-middle panel this is only supported by one data point (arrow). Otherwise, like in the right panel the trace increases gradually; but this is very difficult to judge because of the variability. Same in 4B-middle+right panels. The traces are gradually declining already prior to the onset of the odor down ramp. Moreover, the rates in 4A and 4B right panels are not found to remain consistently above/ below the threshold. I think these experiments require more repetitions, statistics and a no-odor-ramp control of the same larger dataset size. Otherwise the major conclusions of the paper are not sufficiently supported.

5) The differentiation between "reflex" and temporally delayed decision in up versus down ramp is not well founded in the given data. In fact, both responses in 4A vs 4B show very similar profiles, just with opposite signs. The difference is made mainly by the chosen thresholds.

6) The direct functional role of sensory information encoding in directional choice (as stated in the first sentence of the Discussion) is in fact never established due to the open loop configuration. This conclusion should be toned down.

Reviewer #3:

In this study the authors investigate the neural mechanisms underlying nonanone avoidance in *C. elegans*. Using behavioral tracking experiments, they find that, as previously shown for other types of worm chemotaxis, animals show a higher rate of turning when traveling in the aversive direction (here, toward higher repellent concentrations). In addition (and in contrast to salt chemotaxis), the authors find that following these reorientations the angle of bearing is biased away from the source of repellent. Through calcium imaging experiments, they correlate the increase in turn probability with the ASH sensory neurons, which show a rapid on response to nonanone increases, and the bias in run direction to the AWB neurons, which show a slow, "leaky integrator" off response to nonanone decreases. Finally, they analyze a number of signaling and ion channel mutants to explain molecular basis of these differing neural responses.

In general, this is a very interesting paper. The behavioral analysis and subsequent calcium imaging from the respective sensory neurons provides a satisfying explanation for the behavioral strategy underlying aversive chemotaxis and its basic neural mechanism. However, I had a number of questions/concerns regarding the genetic experiments investigating the roles of particular calcium channels in sensory neuron dynamics, and found some of the conclusions here to be overstated. Specifically:

1) The genotypes in all the genetic experiments are poorly and incompletely documented. For example, the alleles used for *egl-19, unc-68, unc-2*, and *itr-1* are not stated (at least I couldn't find them) and it is not stated how many times the strains were backcrossed.

2) From what I can tell, the authors only analyzed a single allele of each gene in question, and did not test whether any were rescued cell-specifically in the cells whose activity was measured. Since most of these genes are pan-neuronal or otherwise broadly expressed in the nervous system, it is not possible to reliably infer that they are functioning cell-autonomously in a particular neuron. Moreover, by only testing a single allele and not testing for rescue, it is not possible to even be sure the effect is due to the gene of interest as opposed to something else in the background.

3) Regarding the calcium channel mutants, the results in ASH are hard to reconcile with the results of Zarhatka et al., 2015. These authors showed that NemA completely blocked cell body calcium transients in ASH in response to a different odorant, octanol. Since presumably both octanol and nonanone are sensed in the cilium, it is hard to explain how the responses they induce could be conveyed to the cell body through different voltage-sensitive channels. Do the authors also see only a partial reduction in octanol response in their system? Is the fast initial response affected by the other (untested) VGCC gene *cca-1*? Also, the effect of the *egl-19* reduction of function mutation is not shown in the figure. How does *egl-19(lf)* compare quantitatively with the supposedly complete block caused by NemA? Obviously it is not necessary for the authors' nonanone results to match Zarhatka's octanol results, but the differences are striking enough to merit further investigation.

4) I also thought the authors slightly overinterpreted the *odr-3* results. *odr-3* is expressed in many olfactory neurons besides AWB, so without cell-specific rescue it is a leap to infer cell-autonomy. Moreover, *odr-3* mutants have been reported by Bargmann et al. to have abnormal cilium structure, so its effect may be of a developmental rather than a signaling nature. More generally, the nature of olfactory G-protein signaling is not well-understood in any *C. elegans* neuron, so inferring a specific role in signal integration on the basis of a single, unrescued allele (and without understanding the signaling basis of the primary response) seems problematic.

In summary, I really like the first part of the paper, but I think the mutant analysis at the end would require a lot of additional genetic experiments (multiple alleles, cell-specific rescue, perhaps testing other candidates like *cca-1* and *gpa-6*) to justify the authors' conclusions. I think it is probably beyond the scope of this paper to do all these extra experiments, so I would instead recommend the authors correct the most important omissions and otherwise scale back their molecular conclusions. Such a paper would still be a very interesting study.

Reviewer #4:

Tanimoto et al. use the well-defined *C. elegans* chemosensory system to probe the relationship between neural activity patterns and behavior. They combine optogenetics with calcium imaging and behavioral analysis show that ASH and AWB sensory neurons use two different strategies to encode odor information. They also link different calcium channels with these two strategies. Importantly, they claim that the different strategies might explain decision making in behavior. There are some interesting data here, but I feel that some of it has been over interpreted. I do think with some changes, this should be a very interesting story.

Major comments

1) What are the differences between "bearing at run initiation" and "bearing after a turn"? Does "bearing after a turn" include the turn with the pirouette? It would be nice to see an example trace to recognize these differences clearly.

2) I believe the authors generated panel Figure 2 by comparing dC/dt in 1 sec window to p(turn) in the two different behavioral phases. They then state 'the efficient transitions between discrete behavioral states based on odor concentration information…'. This quote, along with the surrounding text, seems to imply that odor concentration drives the behavioral transition. Although this is likely true, I see no evidence for it here. For instance, let us consider a worm model in which the worm randomly transitions to a high turning mode (one that is insensitive to C). In this model, the worm would still behave differently at every dC/dt value. Although this model may be silly, it demonstrates that there are models that comply with panel D in which dC/dt does not drive behavioral transitions.

3) I don't understand why the authors used two different types of models for 4A and 4B. The AWB model looks like a more biologically realistic model (at least, it's a standard model). I'm pretty sure the AWB model could be fit to both ASH and AWB. For instance, if we use (1/tau) dx/dt = a*dC/dt – b*x(t), we should be able to use a faster time constant to produce ASH (https://www.wolframalpha.com/input/?i=dx%2Fdt+=+1+-+.1*x) and a slower time constant to produce AWB (https://www.wolframalpha.com/input/?i=10*dx%2Fdt+=+1+-+.1*x). I think it is important to use the same (more biologically relevant) model for both neurons if possible. While this might be beyond the scope, an even more realistic model would probably be: (1/tau) dx/dt = a*f(C) – b*x(t) where f is a function representing receptor saturation.

4) It would be nice if the authors clearly label 'contribution of AWB', etc. in panel 5A. Also, the authors find that the 'time-integral' model produces a more robust/accurate behavior (C). This might give more credence to fitting 'time-integral' models to both AWB and ASH (see comments for F4).

5) The authors used unc-13, but I would also recommend using unc-31 to test whether that this might affect the time delay in AWB.

6) *Odr-3* results are a bit confusing. Given that the nonanone receptor has not been identified, do the authors claim that *odr-3* is coupling to this unknown receptor(s). Or is the property of AWB and ASH modified in an *odr-3* loss of function.

7) I would also recommend using cell-specific knockouts of the calcium channels, which would allow the authors to isolate the effect to the neurons being analyzed.

[Editors' note: further revisions were requested prior to acceptance, as described below.]

Thank you for resubmitting your work entitled "Calcium dynamics regulating the timing of decision-making in *C. elegans*" for further consideration at *eLife*. Your revised article has been favorably evaluated by Eve Marder (Senior editor), a Reviewing editor, and three reviewers.

The manuscript has been improved but before final acceptance, we would like you to address the relatively minor issues raised by Reviewer #2 and Reviewer #4 below.

Reviewer #2:

In the revised version the authors made a good effort to address my concerns. Besides providing additional repetitions for some of their experiments, I requested convincing statistical tests that show whether ASH and AWB imaging data indeed can be better fitted to either one of the models. After their revisions, I am convinced that this is the case for AWB data, which indeed are better explained by the leaky integrator equation. The ASH fitting data however are less convincing. But I agree with the authors that ASH response profiles might be better represented by the time differential during the onset of the response. I think the authors do an appropriate job in addressing this weakness of the study in their discussion, but why leaving future readers with a question mark until they make it to this point. Therefore, I recommend they should give a bit more emphasis on explaining the rather poor fit results for ASH already in the Results section. Allover, I think this paper is a tour de force involving sophisticated technology and analyses. The results are very interesting and provide a working hypothesis for exciting future studies.

Reviewer #3:

I think the authors have done a good job addressing previous reviewer comments. It is an interesting study, and I am happy to recommend publication.

Reviewer #4:

I am satisfied with this version and the authors efforts to address the reviewer's comments.

One additional thing would be to add "saturation of the input" to the standard biophysical model. They could do this by putting the input through a logistic sigmoid or something similar. While this would be nice, it is not essential.

I recommend accepting the manuscript for publication.

---

## [Author Response]

Reviewer #2:

[…]

Major comments

1) Experiments in Figure 1–Figure 2 are performed in a closed loop configuration, i.e. movement directly feeds back onto sensory input. This is not the case for the open loop virtual gradient setup in subsequent experiments. This difference should be better addressed in the discussion text.

We agree with the reviewer's comment. The problem with the traditional behavioral assay is that it is a closed loop, in which the causal relationship between sensory perception and behavior is not necessarily clear. That is why we introduced the robot microscope system. Using this open loop system, we were able to show that a slight difference in sensory input can cause discrete transitions in behavioral states. In order to establish whether the findings in the open-loop system are comparable to the ones in closed-loop systems, we are planning to extend our current open loop system to development of a closed one in the near future. We mention this point in the Discussion section.

2) Figure 4 and E. I am not convinced that AWB and ASH sensory response profiles could not be fitted equally well to the alternative model in each panel. The authors should provide goodness of fit results for all models on each neuron dataset and perform appropriate statistical tests showing that one model performs significantly better than the corresponding null hypothesis model.

In our understanding, the generally used null hypothesis test, which evaluates an absolute goodness of fit, is not well defined for nonlinear models. (For linear models, the null hypothesis would be "no linear relationships".) Thus, we were concerned about misusing null hypothesis testing under inappropriate working assumptions. We think that it is safest to keep our model selection not in an absolute sense but in a relative sense. Therefore, we used the Bayesian information criterion (BIC) to evaluate a relative goodness of fit. (If the reviewer is aware of a suitable null hypothesis for use in nonlinear models, we would be very happy to re-evaluate our tests with it.)

Using BIC, the goodness of fit was higher with the time-integral model than the time-differential model for ASH as well as AWB responses (Table 3). However, at the onset of the ASH response, the integral model exhibited delayed increases compared to the real response (Figure 4—figure supplement 3, red arrows). The overall ASH response appeared to fit better by the combination of the time-integral component plus the fast and transient component as shown in Figure 7, suggesting that ASH has an additional mechanism to quickly respond to a small change in sensory input. However, because the fast and transient component cannot be mathematically expressed at this point, it could not be applied to the result of Figure 4.

If possible, we would like to start the mathematical argument of the ASH response from its fit in the time-differential model, and eventually discuss the fitting in the time-integral model for the following reasons:

1) As described in the original manuscript, the time-differential response is consistent with other recent reports and with the general idea that sensory neurons have phasic properties to sensitively detect temporal changes in stimuli.

2) For general (neuro)biologists, it is more understandable if the mathematical argument starts from a simple time-differential equation before the leaky integrator equation. (At least this is true for the co-authors of this manuscript.)

As for *odr-3*, the fitness was much higher in the leaky integrator equation using *∆t* as a free parameter than that using *∆t* = 1 s (Figure 6—figure supplement 1 and Table 4).

We have mentioned these points in the revised manuscript.

3) The authors use a threshold to determine when sensory information leads to an abrupt change in turning frequency. It is not sufficiently explained how exactly this 99%-prediction-interval is calculated. Provide more information in main text about the logic and details in Materials and methods section.

We apologize that the initial explanation was not sufficient. The 99% prediction interval is a value where future data (odor-up or odor-down phase in this case) will fall with 99% probability based on the observed data (odor-zero or plateau phase, respectively). Using this criterion, we analyzed timing when the unexpected value appeared for the first time in Figure 4.

We did this because we intended to investigate the *timing* of turning rate's change after we have found that the turning rate in the odor-up and the odor-down phases significantly increased and decreased, respectively (Figure 3). In order to investigate the timing, it was necessary to calculate the changes in turning rate in narrower temporal windows. However, since turning is a rare occurrence, narrowing the time window increases the variation; in order to obtain the same turn number as in Figure 3 (30 or 90 seconds) at the time window of 10 seconds, sampling had to be performed 3 or 9 times. Furthermore, considering comparisons among multiple temporal windows, the use of a regular statistical analysis is not practical. Therefore, we used prediction interval, based on the idea of statistical inference. We described the logic and the calculation in the Results and Materials and methods section, respectively.

4) I am not convinced that the turning rate profiles in Figure 4 middle and right panels show discrete transitions. In Figure 4-middle panel this is only supported by one data point (arrow). Otherwise, like in the right panel the trace increases gradually; but this is very difficult to judge because of the variability. Same in 4B-middle+right panels. The traces are gradually declining already prior to the onset of the odor down ramp. Moreover, the rates in 4A and 4B right panels are not found to remain consistently above/ below the threshold. I think these experiments require more repetitions, statistics and a no-odor-ramp control of the same larger dataset size. Otherwise the major conclusions of the paper are not sufficiently supported.

We appreciate the reviewer's comments. We believe the difference has been made clearer as a result of the additional experiments. In the case of ASH, turning rates in the 45 s and 90 s odor-up conditions (the left-most and middle left panels, respectively) were consistently higher than the prediction interval for the odor-zero phase and were significantly higher than in the no-odor control in the first 10-s bin of the odor-up phase in multiple comparison tests (Figure 4 and [Supplementary-material SD1-data]). Although the turning rate in the 180 s odor-up condition (middle right) in the first bin was not statistically different, the rate in the odor-up phase was also above the 99% prediction interval during the first 90 s; the following decrease may possibly be due to adaptation.

For the AWB response in Figure 4, the 90 s odor-down condition (middle left) exhibited a step-like change in turning rate, although its timing was delayed for 10 s compared to the 45 s condition (left-most). In the 180 s odor-down condition (middle right), we showed that turning intervals were still classified to high- and low-turning states (Figure 1—figure supplement 1). This result indicates that discrete behavioral transitions occurred at different timings between individuals, resulting in the gradual change of tuning rate as a group. We also found that all the timings passing the 99% prediction interval were associated with statistical differences compared to odor-plateau control ([Supplementary-material SD1-data]).

We added statistical arguments and modified the sentences accordingly (subsection “Temporal differentiations and integration of odor information regulate avoidance behavior”).

5) The differentiation between "reflex" and temporally delayed decision in up versus down ramp is not well founded in the given data. In fact, both responses in 4A vs 4B show very similar profiles, just with opposite signs. The difference is made mainly by the chosen thresholds.

In our experimental conditions, the turning rate changed from a low to a high state (or *vice versa*), so the overall profiles were similar. Rather, the relevant question is the *timing* of said changes. As described above, a substantial behavioral change occurred at the onset of the odor concentration increases in all of the three conditions, including the 180 s odor-up condition (middle right) which causes a near-threshold-level behavioral response. In contrast, the behavioral response to the odor concentration decreases occurred at different timings in three conditions according to the rate of decrease. In our view, these results suggest that increasing and decreasing odor concentrations cause a discrete behavioral state transition, but their timings are statistically different, which is related to the dynamic activities of ASH and AWB neurons. These results are consistent with our original claims, and we therefore did not think it necessary to change the sentence.

6) The direct functional role of sensory information encoding in directional choice (as stated in the first sentence of the Discussion) is in fact never established due to the open loop configuration. This conclusion should be toned down.

We have toned this conclusion down according to the suggestion.

Reviewer #3:

[…]

1) The genotypes in all the genetic experiments are poorly and incompletely documented. For example, the alleles used for egl-19, unc-68, unc-2, and itr-1 are not stated (at least I couldn't find them) and it is not stated how many times the strains were backcrossed.

We apologize that the allele names were written only in the strain table in the Materials and methods section, and that the original strain names were not described anywhere by mistake. The strains are mostly reference alleles, and all of them were used in previous studies (see below). We have added the strain names and allele names in the Materials and methods section.

2) From what I can tell, the authors only analyzed a single allele of each gene in question, and did not test whether any were rescued cell-specifically in the cells whose activity was measured. Since most of these genes are pan-neuronal or otherwise broadly expressed in the nervous system, it is not possible to reliably infer that they are functioning cell-autonomously in a particular neuron. Moreover, by only testing a single allele and not testing for rescue, it is not possible to even be sure the effect is due to the gene of interest as opposed to something else in the background.

We also apologize that the data was not clearly or sufficiently provided in the initial version of the manuscript. First, 2 alleles were used for *odr-3* (one allele was presented in the Supplementary Figure). As for other mutants, the strains used here have been investigated for ASH response in previous studies, mostly as single alleles without rescue (Hilliard et al., 2005; Kato et al., 2014; Pokala et al., 2014; Zahratka et al., 2015). In addition, the effects of *egl-19* or *unc-68* genetic mutations were similarly reproduced by pharmacological antagonists, supporting the idea that the effects were due to the suppression of gene products. (The ASH response under treatment with dantorolene, a RyR antagonist, was added in Figure 7; it was not presented in the initial submission.) Moreover, the use of an antagonist is more advantageous than using two alleles in one particular way: Pharmacological treatment ensures that the gene product is required in the developed nervous system, not in the developing one. We corrected the sentences accordingly (subsection “Molecular mechanisms of neural computation in the sensory neurons”).

Cell-autonomy: Since both AWB and ASH are sensory neurons and *unc-13* mutation, which abolishes synaptic transmission, did not essentially affect the neuronal responses (ASH data was also added in Figure 7—figure supplement 1), we think it is reasonable to consider that the effects of genes on calcium responses are cell-autonomous.

Taken together, we consider that our main arguments that "ODR-3 and EGL-19 may play important roles in AWB" and "CICR by RyR may also play an important role together with EGL-19 in ASH" are sufficiently supported by the data.

Still, we agree that the involvement of these gene products in ASH/AWB responses were not demonstrated directly. Thus, we toned down the offending sentences.

3) Regarding the calcium channel mutants, the results in ASH are hard to reconcile with the results of Zarhatka et al,. 2015. These authors showed that NemA completely blocked cell body calcium transients in ASH in response to a different odorant, octanol. Since presumably both octanol and nonanone are sensed in the cilium, it is hard to explain how the responses they induce could be conveyed to the cell body through different voltage-sensitive channels. Do the authors also see only a partial reduction in octanol response in their system? Is the fast initial response affected by the other (untested) VGCC gene cca-1? Also, the effect of the egl-19 reduction of function mutation is not shown in the figure. How does egl-19(lf) compare quantitatively with the supposedly complete block caused by NemA? Obviously it is not necessary for the authors' nonanone results to match Zarhatka's octanol results, but the differences are striking enough to merit further investigation.

We agree that the results of Zahratka et al. appear different from ours. However, it may be difficult to conclude that there is no transient response at all in their experiment because they did not display the temporal profile of the response in the relevant figure. (They showed temporal profiles in other figures, though.) Furthermore, high osmolality stimulation, which also causes calcium response in ASH neurons and avoidance behavior, was not substantially affected by the *egl-19* mutation (Kato et al., 2014; Pokala et al., 2014). Thus, these results were completely different although 1-octanol and glycerol were both similarly presented in the aqueous phase in a stepwise manner in these experiments. Because the polymodal ASH neurons respond to a wide range of aversive signals, they may respond to different stimuli with different mechanisms. Still, we agree that the differences are very interesting, and we are planning to investigate this in our next project.

We analyzed the T-type VGCC *cca-1* mutant and found that the ASH response was not affected (Figure 7).

The loss-of-function allele of *egl-19* is not available, possibly due to lethality. NemA-treatment showed a similar or stronger effect than the reduction-of-function *egl-19* mutations in the previous studies (Kwok et al., 2006; Zahratka et al., 2015), and a consistent result was observed in AWB neurons in our study (Figure 7). Thus, we did not analyze the *egl-19* reduction-of-function mutation in ASH neurons.

We added relevant sentences to the manuscript (“Molecular mechanisms of neural computation in the sensory neurons”)

4) I also thought the authors slightly overinterpreted the odr-3 results. odr-3 is expressed in many olfactory neurons besides AWB, so without cell-specific rescue it is a leap to infer cell-autonomy. Moreover, odr-3 mutants have been reported by Bargmann et al. to have abnormal cilium structure, so its effect may be of a developmental rather than a signaling nature. More generally, the nature of olfactory G-protein signaling is not well-understood in any C. elegans neuron, so inferring a specific role in signal integration on the basis of a single, unrescued allele (and without understanding the signaling basis of the primary response) seems problematic.

As in our response to the second comment, because the AWB response was essentially not affected by the *unc-13* mutation, we think it is reasonable to consider that the AWB response is cell-autonomous. In addition, the effect of the *odr-3* mutation was analyzed in 2 alleles in the original version of the manuscript (mentioned above). These results suggest that the different AWB responses are likely due to the lack of ODR-3 activity in the neurons.

The response of AWB neurons in *odr-3* mutants is even larger than that in wild-type animals. This could be due to a developmental effect, as suggested by the reviewer. Still, our mathematical analysis (which was possible because we were able to precisely control the odor concentration input) suggested one possibility: ODR-3 may be involved in the time-differential computation. This idea has never been proposed. We believe that this interesting possibility should be reported to the readers, but agree that more experiments are needed to fully prove our contention. Therefore, we toned down the related sentences.

Reviewer #4:

[…]

Major comments

1) What are the differences between "bearing at run initiation" and "bearing after a turn"? Does "bearing after a turn" include the turn with the pirouette? It would be nice to see an example trace to recognize these differences clearly.

Sorry for the confusion. In Figure 1, the bearing at run initiation is the direction of the root of the blue long arrow, and the bearing after a turn is the direction of the root of the blue long arrow and of the red short arrow. We added an explanatory sentence (Figure 1 legend).

2) I believe the authors generated panel Figure 2 by comparing dC/dt in 1 sec window to p(turn) in the two different behavioral phases. They then state 'the efficient transitions between discrete behavioral states based on odor concentration information…'. This quote, along with the surrounding text, seems to imply that odor concentration drives the behavioral transition. Although this is likely true, I see no evidence for it here. For instance, let us consider a worm model in which the worm randomly transitions to a high turning mode (one that is insensitive to C). In this model, the worm would still behave differently at every dC/dt value. Although this model may be silly, it demonstrates that there are models that comply with panel D in which dC/dt does not drive behavioral transitions.

As pointed out in minor comment 3 of reviewer # 2, the causality is not known at this stage, and our argument is only one of the possibilities. We made this clear in the text (Results section).

3) I don't understand why the authors used two different types of models for 4A and 4B. The AWB model looks like a more biologically realistic model (at least, it's a standard model). I'm pretty sure the AWB model could be fit to both ASH and AWB. For instance, if we use (1/tau) dx/dt = a*dC/dt – b*x(t), we should be able to use a faster time constant to produce ASH (https://www.wolframalpha.com/input/?i=dx%2Fdt+=+1+-+.1*x) and a slower time constant to produce AWB (https://www.wolframalpha.com/input/?i=10*dx%2Fdt+=+1+-+.1*x). I think it is important to use the same (more biologically relevant) model for both neurons if possible. While this might be beyond the scope, an even more realistic model would probably be: (1/tau) dx/dt = a*f(C) – b*x(t) where f is a function representing receptor saturation.

Please refer to the response to major comment 2 of reviewer #2 (the 2nd and following paragraphs).

4) It would be nice if the authors clearly label 'contribution of AWB', etc. in panel 5A. Also, the authors find that the 'time-integral' model produces a more robust/accurate behavior (C). This might give more credence to fitting 'time-integral' models to both AWB and ASH (see comments for F4).

According to the suggestion, we modified Figure 5. For ASH's "time-integral" model, please refer to the comment above.

5) The authors used unc-13, but I would also recommend using unc-31 to test whether that this might affect the time delay in AWB.

According to the suggestion, we analyzed the unc-31 mutants, required for the priming of dense core vesicles, and found that the response also fit the leaky integrator equation. We added the data in Figure 6 and mentioned this in the text (subsection “Temporal integration of sensory information occurs in AWB neurons”).

6) Odr-3 results are a bit confusing. Given that the nonanone receptor has not been identified, do the authors claim that odr-3 is coupling to this unknown receptor(s). Or is the property of AWB and ASH modified in an odr-3 loss of function.

We assume that reviewer # 4’s concern is whether ODR-3 Gα is more likely activated by an odorant receptor for sensory signal transduction, or by a neuromodulator receptor to regulate the response of AWB and ASH. From genetic studies so far, the GPCR -> Gα -> cGMP -> cGMP-gate cation channel pathway is considered to be activated in AWB neurons (Bargmann, 2006). In addition, ODR-3 is expressed in only a few pairs of sensory neurons, and localizes at the sensory ending (*i.e.*, sensory cilia), suggesting that ODR-3 is involved in sensory signaling (Bargmann, 2006). On the other hand, other Gαs, such as GOA-1, are expressed in many neurons and have been suggested to function downstream of neuromodulator receptors (Bastiani and Mendel, 2006). We added sentences in this regard (subsection “Molecular mechanisms of neural computation in the sensory neurons”).

7) I would also recommend using cell-specific knockouts of the calcium channels, which would allow the authors to isolate the effect to the neurons being analyzed.

As mentioned in the response to major comment 2 of reviewer # 3, because the cell responses were essentially not affected by the *unc-13* mutation, and because they are sensory neurons, we believe it is reasonable to think the effect is cell-autonomous.

[Editors' note: further revisions were requested prior to acceptance, as described below.]

Reviewer #2:

In the revised version the authors made a good effort to address my concerns. Besides providing additional repetitions for some of their experiments, I requested convincing statistical tests that show whether ASH and AWB imaging data indeed can be better fitted to either one of the models. After their revisions, I am convinced that this is the case for AWB data, which indeed are better explained by the leaky integrator equation. The ASH fitting data however are less convincing. But I agree with the authors that ASH response profiles might be better represented by the time differential during the onset of the response. I think the authors do an appropriate job in addressing this weakness of the study in their discussion, but why leaving future readers with a question mark until they make it to this point. Therefore, I recommend they should give a bit more emphasis on explaining the rather poor fit results for ASH already in the Results section. Allover, I think this paper is a tour de force involving sophisticated technology and analyses. The results are very interesting and provide a working hypothesis for exciting future studies.

We appreciate the reviewer's positive feedback and suggestion. We have added a sentence to provide the explanation of the ASH fitting ASH to the Results section.

Reviewer #4:

I am satisfied with this version and the authors efforts to address the reviewer's comments.

One additional thing would be to add "saturation of the input" to the standard biophysical model. They could do this by putting the input through a logistic sigmoid or something similar. While this would be nice, it is not essential.

I recommend accepting the manuscript for publication.

We appreciate the suggestion. We have added a sentence regarding the consideration of input saturation to the Materials and methods section.